# On the Low-Rank Parametrization of Reward Models for Controlled Language Generation

**Sergey Troshin**                                    *s.troshin@uva.nl*
*Language Technology Lab, Informatics Institute, University of Amsterdam*

**Vlad Niculae**                                      *v.niculae@uva.nl*
*Language Technology Lab, Informatics Institute, University of Amsterdam*

**Antske Fokkens**                                    *antske.fokkens@vu.nl*
*Computational Linguistics and Text Mining Lab,*
*Faculty of Social Sciences and Humanities, Vrije Universiteit Amsterdam*

**Reviewed on OpenReview:** *https://openreview.net/forum?id=cjRsEGLT8B*

## Abstract

Language models trained on large amounts of data are known to produce inappropriate content in some cases and require careful tuning to be used in the real world. We revisit an effective and modular approach for controllability of the language models, when an external expert model guides the decoding. Particularly, we zoom in into the *parametrization choice* of an external expert, highlighting the difference between low-rank and higher-rank parametrizations. Higher-rank experts are designed to support high flexibility when representing the rewards, leading to higher computational costs during decoding. However, we demonstrate that they might not use their full flexibility. By analyzing the recently proposed reward-augmented decoding approach (RAD), which uses a higher-rank expert model, we introduce a simpler but more efficient low-rank parametrization of the expert model enabling fast and effective guided decoding. We empirically show that the low-rank RAD performs on par with the more flexible RAD on a detoxification and a sentiment control task, while requiring only a single reward model call per generated token.

## 1 Introduction

Generative large language models (LLMs) have gained a lot of popularity in recent years and shown impressive results in zero-shot and few-shot scenarios on numerous downstream tasks (Touvron et al., 2023; OpenAI, 2024; Jiang et al., 2023). These large-scale models are pretrained on large amounts of data, and are known to inherit and memorize underlying biases (Sheng et al., 2019) as well as to provide unsafe responses (Wallace et al., 2019; Ganguli et al., 2022), necessitating further tuning for safer deployment and control (Ouyang et al., 2022).

Control over LLMs can be roughly divided into methods which modify the original model via finetuning (Ouyang et al., 2022; Rafailov et al., 2023), and decoding-time solutions, which do not modify the parameters of the original model, including best-of-n sampling (Wang et al., 2023; Sun et al., 2024). As models increase in size, finetuning becomes prohibitive with limited computational resources. In this work, we focus on a more modular approach of decoding-time guidance, and assume we have access to top-$k$ logits of a black-box base language model (see §2.1 for details). In this line of work, a discriminator model is trained to modify or re-rank the logits of the base model during decoding in order to satisfy the desired constraint (Yang & Klein, 2021), while preserving the distribution of the language model as much as possible.

In our work, we zoom in on the parametrization choice of the reward model. Yang & Klein (2021), Deng & Raffel (2023), Mudgal et al. (2024), Chakraborty et al. (2024) parameterize reward models as discriminators,

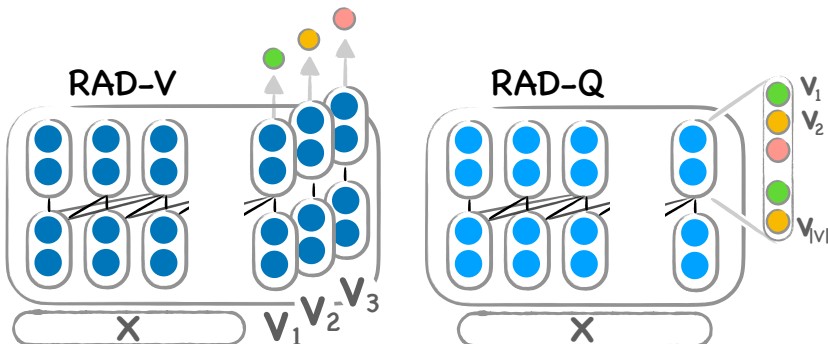

**Figure 1:** The RAD-V parametrized model (left) predicts a reward for each next token candidate $v$ independently concatenating it to the current prefix $x$. RAD-Q (right) parametrization uses the language model output embeddings to efficiently predict the rewards for next token candidates over the vocabulary $V$.

which requires a separate forward pass through the backbone of the reward model (layers before the output head) for each token candidate. Liu et al. (2021), Krause et al. (2021) and Cao et al. (2022) use more efficient approaches and perform a single forward pass to predict the scores for all next token candidates. A similar parametrization choice is encountered in the reinforcement learning literature (RL), where one chooses between state value function ($V$ function) or state-action value function ($Q$ function) parametrization (Sutton & Barto, 2018; Dulac-Arnold et al., 2015). In particular, closest to our work are the decoding-time RL approaches, where V/Q functions are trained to guide the decoding from a language model (Mudgal et al., 2024; Chakraborty et al., 2024). In our work, we highlight the rank bottleneck (Yang et al., 2018) of the $Q$-style parametrized models in application to text reward modeling, where vocabulary size is larger than model dimension. We theoretically analyze the difference between these two parametrizations in terms of rank expressivity. We provide a simple but realistic example, where $Q$-style parametrization is less expressive compared to $V$-style parametrization, suggesting that the $Q$-style choice of parametrization should not be treated as the default option indiscriminately.

Despite the rank bottleneck limitation, $Q$-style parametrization is more computationally attractive. We therefore ask whether there are cases where it is justifiable to use $Q$-style parametrization, which we answer affirmatively. Particularly, we look at the two popular controlled generation tasks and a recently proposed reward augmented decoding (RAD) approach (Deng & Raffel, 2023), where they train a $V$-style parametrized model (we will refer to it as RAD-V). While RAD-V demonstrates high effectiveness for controlled generation, it scales poorly when the number of next token candidates grows, requiring a separate forward pass through the backbone of the reward model for each token candidate. We demonstrate that for this scenario, high rank is not required to approximate the training data well, and hence RAD-V might not use its full flexibility. To empirically verify that a RAD approach using $Q$-style parametrization, RAD-Q, is then expressive enough for this scenario, we distill RAD-V into an efficient RAD-Q, and demonstrate that guided decoding with the RAD-Q model results in a comparable attribute control/fluency to the more flexible but more computationally intensive RAD-V approach.

## 2 Preliminaries

### 2.1 Guided decoding with external experts

In this section, we outline the approach of guiding a base language model with external token-level discriminators. At each step of decoding, both the base model and the discriminator observe an already generated prefix $x$, and cooperate to score the next token candidates $v \in V$. A language model predicts the logits $z_{\mathrm{LM}}(\cdot|x) \in \mathbb{R}^{|V|}$ and the goal of discriminator is to augment these logits with reward scores $\hat{r}(\cdot|x) \in \mathbb{R}^{|V|}$. A standard practice is to consider only likely tokens $V' \subseteq V$ at each decoding step *e.g.* via top-$k$ (Fan et al.,

2018; Deng & Raffel, 2023) or nucleus sampling (Holtzman et al., 2020):

$$z(v|x) = \begin{cases} z_{\text{LM}}(v|x) + \beta \hat{r}(v|x), & \text{if } v \in V', \\ -\infty, & \text{otherwise,} \end{cases} \tag{1}$$

and the next token is sampled from the categorical distribution:

$$\tilde{p}(x) = \text{Softmax}(z(v|x)). \tag{2}$$

While some language models might have a restrictive application programming interface (API) for safety reasons, this line of work makes a reasonable assumption that we have access to the top-$k$ logits of a language model either directly or via API for a relatively small $k \ll |V|$.

To define reward scores, $Q$-style models given a prefix $x$ only pass it once through the external language model backbone, and use the linear output layer to obtain the scores for each of the next token candidates. GeDi (Krause et al., 2021) and DExperts (Liu et al., 2021) use attribute-conditioned unidirectional language models (undesired attribute in GeDi or two LM experts for desired and undesired attribute in DExperts), trained via the standard language modeling objective on class-conditioned data: $\hat{r}_y(v|x) = z_t(v|x, y)$, where $y \in \{0; 1\}$ is the attribute (*e.g.* positive/negative sentiment).

Alternatively, a $V$-style parametrized model such as RAD-V (Deng & Raffel, 2023) predicts the attribute of interest for a prefix *concatenated* with a next token candidate $\hat{r}_{\text{RAD-V}}([x, v])$, where $[\cdot, \cdot]$ denotes the concatenation of a prefix and a next token candidate. This approach requires passing each next token candidate as *input* to the model, thus, to obtain the scores for $k$ next token candidates $v$ for top-$k$ decoding RAD-V would need $k$ forward calls of the reward model, which can slow down inference significantly and constrains them to limit the number of next token candidates. Therefore, we ask whether $Q$-style parametrized models can possibly match the performance of $V$-style models while enjoying higher efficiency.

### 2.2 RAD training

In this section, we outline how the RAD approach (Deng & Raffel, 2023) uses labeled data to train a reward model.

At the training stage, we assume that we have a dataset $\mathcal{D} = \{(u^{(i)}, y^{(i)})\}_{i=1}^{n}$ of $n$ text utterances $u$ of length $l(u)$ and responses $y \in [0; 1]$. The RAD approach is to use the data distribution to estimate the expected future response. For each utterance $u$ from the dataset, we can split it prefix $x'$, next token $v$, and a continuation $\mathbf{v}$, in all possible ways. This way, they create an extended dataset to train on partial prefixes:

$$\mathcal{D}_f = \{(x, u, y \mid x = u_{1:t}, t \in (1, \dots, l(u)), (u, y) \in \mathcal{D}\}. \tag{3}$$

RAD trains a reward model to predict $y$ given a text input. Then, during training, RAD takes the input prefix $x = [x', v]$ and incurs a weighted squared loss for approximating the future reward:

$$\mathcal{L}(\hat{r}(v|x'), y, \lambda) = \lambda \cdot (\hat{r}(v|x') - y)^2, \tag{4}$$

where $\lambda$ are the discounting weights $\lambda(x, u) = l(x)/Z_u$, for each prefix used to up-weight prefixes closer to the full sentence, and $Z_u = \sum_{t=1}^{l(u)} l(u_{:t})$ is a normalizing constant such that $\sum_{t=1}^{l(u)} \lambda(u_{:t}, u) = 1$. During training, we can use teacher forcing to process all prefixes of an utterance in a single pass.

## 3 Reward modeling as low-rank matrix factorization

### 3.1 Analysis of RAD

#### 3.1.1 Reward modeling as matrix completion

To better understand the training objective of RAD, we start by looking at the optimization problems defined in Equation (4), where we optimize a reward model to approximate future responses. A unidirectional reward

model can predict a reward value for each next next token candidate. If we enumerate all the contexts $x'$ in the training data and all possible next tokens $v$, we task a reward model to predict the values of $R \in \mathbb{R}^{N \times |V|}$, which we dub the *reward matrix*.

If each context would be observed only once, $R$ would have a single observed reward in each row. For short and common contexts we can observe more continuations per row, and also for some contexts there can be ambiguities: $\{(x, u_1, y_1), ...(x, u_m, y_m)\}$. From a mean squared error point of view, it is equivalent to compress these ambiguities by taking the weighted average of their $y$ (Appendix B):

$$R[x', v] = \frac{\sum_{u,y \sim \mathcal{D}_f[x]} \lambda(x, u)\, y}{\sum_{u,y \sim \mathcal{D}_f[x]} \lambda(x, u)}. \tag{5}$$

From this perspective, reward modeling can be interpreted as a matrix completion problem. The training dataset $\mathcal{D}_f$ gives us only an incomplete view of a true reward matrix $R$. Following the notation in the matrix completion literature (Mazumder et al., 2010), $\Omega$ denotes the set of indices of the observed entry indices $\{(x', v) \,|\, x = [x', v], x \in D_f\}$, and $P_\Omega(R)$ denotes the projection of $R$ that sets all indices outside $\Omega$ to zero. We denote the complement of $\Omega$ w.r.t. the complete set of indices as $\neg\Omega$. The full RAD objective for a $V$-style parametrized model is equivalent to minimizing $\|P_\Omega(R) - P_\Omega(\hat{R}_{\text{RAD-V}})\|_F^2$, where each entry $\hat{R}_{\text{RAD-V}}[x', v] = \hat{r}(v|x')$ can be computed with a forward pass.

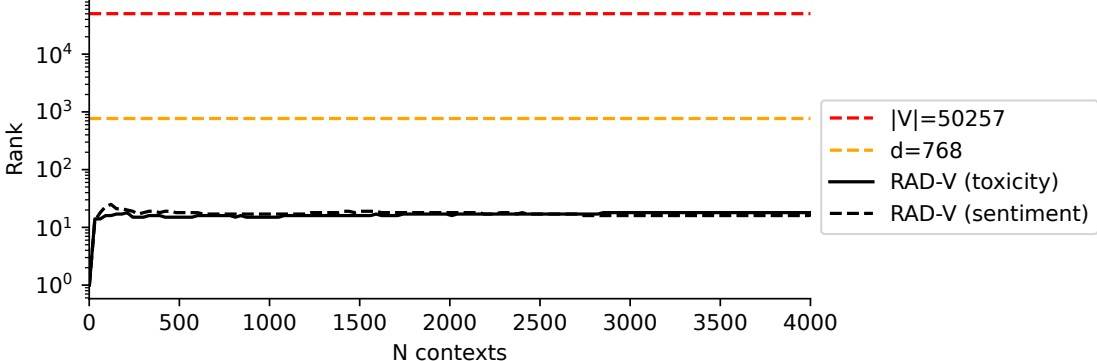

**Figure 2:** When considering a RAD-Q approach *e.g.* distilling RAD-V into RAD-Q, it is important to ask whether the rank capacity of the RAD-Q model is enough to model RAD-V outputs. We numerically estimate the rank of $\hat{R}_{\text{RAD-V}}$ by incrementally considering more rows of the $\hat{R}_{\text{RAD-V}}$ matrix, and observe that the rank tends to be less than the model dimension $d = 764$ and much less than $|V|$, the maximal possible rank of $P_\Omega(R)$.

To analyze the reward matrices from the perspective of rank expressivity, we define the *minimal rank* of a partially observed matrix

**Definition 1.** Let $R$ be a $n \times m$ matrix, with $\Omega$ denoting a set of observed indices. We define the minimal rank of $R$ *w.r.t.* $\Omega$ as the rank of the lowest-rank completion consistent with $P_\Omega(R)$:

$$\min \text{rank}_\Omega(R) := \min \left\{ \text{rank}(\hat{R}) : \hat{R} \in \mathbb{R}^{n \times m}, P_\Omega(R) = P_\Omega(\hat{R}) \right\} \tag{6}$$

Additionally, since numerical solutions to low-rank problems must incur some error, we also define a *minimal numerical rank*.

**Definition 2.** Let $R$ be a $n \times m$ matrix, with $\Omega$ denoting a set of observed indices, and $\varepsilon \in \mathbb{R}_+$. We define the minimal $\varepsilon$-rank of $R$ $w.r.t.$ $\Omega$ as:

$$\min \text{rank}_{\Omega, \varepsilon}(R) := \min \left\{ \text{rank}(\hat{R}) : \hat{R} \in \mathbb{R}^{n \times m}, \frac{1}{nm} \| P_\Omega(R) - P_\Omega(\hat{R}) \|_F^2 \leq \varepsilon \right\}. \tag{7}$$

### 3.1.2 RAD-V can be high-rank, but is not in practice

Given a prefix $x$, RAD-V accepts a token candidate $v$ as an additional **input** to the model $\hat{R}_{\text{RAD-V}}[x', v] = \hat{r}_{\text{RAD-V}}([x', v])$, passing $v$ through the layers of the reward model. For this reason, we expect RAD-V to have the capacity to represent a large space of reward matrices including matrices with higher rank. In Appendix D.1, we empirically verify that **RAD-V is capable to model data that has high minimal rank**, *i.e.*, it can fit (with training loss numerically close to zero) a dataset $R$ that has min $\text{rank}_\Omega(R) > d$, where $d$ is the dimensionality of the Transformer model. This flexibility does come at a cost: to score many next token candidates during top-$k$ decoding, RAD-V requires a forward pass through all layers of the model for each of the $k$ next token candidates. Following this observation, an important question is *do we need this flexibility at the cost of slower decoding?*

In Figure 2, we aim to measure the rank of $\hat{R}_{\text{RAD-V}}$ for RAD-V trained on two datasets: for detoxification and sentiment control tasks (discussed in detail in §4). To numerically estimate the rank, we follow Finlayson et al. (2024) and first sample $N$ random prefixes $x$ from the dataset $\mathcal{D}_f$ to calculate $N$ full rows of $\hat{R}_{\text{RAD-V}}$ (requiring $N \cdot |V|$ calls to the RAD-V reward model). Then we use singular value decomposition with the standard singular value cutoff to compute the *numerical rank* (Appendix D.4). **We observe that the reward matrix learned by RAD-V tends to have *low numerical rank***, suggesting that it is possible to use less flexible but faster reward models to improve the efficiency of reward models.

### 3.1.3 RAD training data is low-rank.

One possible explanation why RAD-V does not use its full flexibility is that said flexibility is not required to fit the training data, *i.e.*, that $P_\Omega(R)$ can be fit with the low-rank model. To analyze the rank needed to fit the data, we use the definition of the minimal numerical rank (Definition 2). Empirically calculating the minimal numerical rank of the data is challenging due to the very large number of prefixes. We use a combination of theoretical and empirical approaches listed in Appendix C.3, and we claim that the RAD training data (Jigsaw (cjadams et al., 2019) and Amazon Polarity (Zhang et al., 2015)) **has low minimal numerical rank**, that is less than the model dimension for $\varepsilon = 10^{-6}$, and even less than 256 if we allow $\varepsilon = 10^{-3}$. We further confirm this claim for the commonly used HelpSteer (helpfulness) dataset (Wang et al., 2024b) and BeaverTails (safety) dataset (Ji et al., 2023), used for reward model training (Wang et al., 2024a). We thus draw the conclusion that the incomplete $P_\Omega(R)$ matrix can be fit with the low-rank matrix factorization with a small error.

### 3.1.4 Is low-rank enough to model reward matrices?

When viewed from a matrix completion angle, it may seem that the reward modeling problem is well suited for low-rank modeling. However, **we will argue this should not be taken for granted**.

First, it is indeed true that data matrices generated by a certain kind of random process (Udell & Townsend, 2019) are low-rank with high probability. Moreover, in the case of missing values, intuition suggests that they should tend to low-rank, and indeed the following result demonstrates that we can almost always decrease the rank by 1 when filling in one missing value (Appendix C.2):

**Lemma 1.** For $k > 0$, let $R$ be a random matrix with distribution supported on $\mathbb{R}^{k \times k}$, and a single missing value, *i.e.*, $\neg\Omega = \{(i, j)\}$. Then, $P(\min \text{rank}_\Omega(R) < k) = 1$.

Moreover, when most values are missing, rank again seems likely to be low. In particular, when only one value is observed per row (as demonstrated in Example 1), a rank-1 completion is possible (Example 2,

Appendix C.1). In a dataset, there will likely be many unique prefixes where this scenario is applicable, particularly for longer prefixes.

> **Example 1.** As illustrated in Figure 3a, assume that all prefixes $x$ appear only once in the dataset, so that $|\{(i,j) \in \Omega : i = x\}| = 1$ for all $x$. Then, $\min \text{rank}_\Omega(R) = 1$.

> **Example 2.** As illustrated in Figure 3b, let each token in the vocabulary either appear in high-reward ($y{=}1$) or low-reward ($y{=}0$) utterances (independent of context). This gives a rank-1 completion consistent with Example 1.

Given this observation, one might assume that having more missing values in a reward matrix always implies that the reward matrix can be fit with lower rank. However, this is not true, and the following result suggests that a reward matrix with many missing values and can still be forced to have a high rank:

> **Lemma 2.** For $k > 0$, there exist $R \in \mathbb{R}^{k \times k}$ and $\Omega$ with $|\Omega| \in O(k^2)$ such that $\min \text{rank}_\Omega(R) = k$.

*Proof.* Consider a $k$-by-$k$ identity matrix with $k(k-1)/2$ missing values above the diagonal (see Figure 3c from Example 3). Any possible completion results in an upper-triangular matrix. The determinant of such a matrix is the product of its diagonal elements, *i.e.*, 1, so any completion must be full rank. □

The construction in Lemma 2 is not artificial and could indeed correspond to a reward modeling task for the context-dependent constraints, as shown in the next examples.

> **Example 3.** Consider a list of (first name, last name) pairs: $[(k_i, v_j)]$, $1 \le i, j \le n$, where some pairs are marked as allowed or blocked. Assume that the rule is as follows: for any utterance containing $(k_i, v_j)$ with $j \le i$, we assign the reward 0; if $j = i$, we assign the reward 1, and unknown otherwise (Figure 3c). We assume that each utterance contains a single mention of a person, then the expected reward matrix with $[\dots, k_i]$ indexing rows and $v_j$ indexing columns will contain the upper-triangular submatrix, which is of rank $n$.

> **Example 4.** Consider an application of a LM to an arithmetical reasoning task, where you have examples in form of $x + y = z$, where $x$, $y$, $z$ are integer number tokens forming a vocabulary $V = [0..999]$. Let us define a reward function as $R(x + y = z) = 1$ if the expression is true, 0 if false. Then by considering all possible contexts $x + y =$, it is easy to show that the reward matrix is full-rank since the rows of the reward matrix include all one-hot vectors representing the correct answer.

Empirically, our results find that Jigsaw (cjadams et al., 2019) and Amazon Polarity (Zhang et al., 2015) indeed support low-rank completions (§4). Nevertheless, from all results in this section, we conclude that care must be taken before making low-rank assumptions about reward datasets.

## 3.2 Low-Rank Autoregressive Reward Model

In this section, we introduce RAD-Q (Figure 1), a low-rank parametrization of RAD, designed for efficient modeling of reward scores for next token candidates.

We revisit the language modeling ($Q$-style) parametrization for reward prediction (Liu et al., 2021; Krause et al., 2021) and predict the scores for all next token candidates with a single forward pass through the backbone of a language model. In contrast to RAD-V, RAD-Q predicts the representation vector $h(x) \in \mathbb{R}^d$ given a prefix $x$ and uses output embeddings $e(v) \in \mathbb{R}^d$ to get the scores for all next token candidates. We

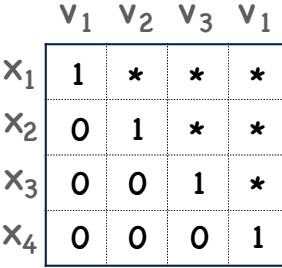

|   | $v_1$ | $v_2$ | $v_3$ | $v_1$ |
|---|---|---|---|---|
| $x_1$ | * | * | 0 | * |
| $x_2$ | * | 0.5 | * | * |
| $x_3$ | 1 | * | * | * |
| $x_4$ | * | * | * | 1 |

**(a)** Sparse $P_\Omega(R)$

|   | $v_1$ | $v_2$ | $v_3$ | $v_1$ |
|---|---|---|---|---|
| $x_1$ | 1 | 0.5 | 0 | 1 |
| $x_2$ | 1 | 0.5 | 0 | 1 |
| $x_3$ | 1 | 0.5 | 0 | 1 |
| $x_4$ | 1 | 0.5 | 0 | 1 |

**(b)** Context independent $R$.

|   | $v_1$ | $v_2$ | $v_3$ | $v_1$ |
|---|---|---|---|---|
| $x_1$ | 1 | * | * | * |
| $x_2$ | 0 | 1 | * | * |
| $x_3$ | 0 | 0 | 1 | * |
| $x_4$ | 0 | 0 | 0 | 1 |

**(c)** Full rank $P_\Omega(R)$.

**Figure 3:** (a) each row of a reward matrix $P_\Omega(R)$ has only 1 known element, the minimal rank is 1; (b) a reward matrix has rank 1, when each token has a context independent reward; (c) a reward matrix is full rank for highly context dependent rewards e.g. for the case of identity matrix.

use the following RAD-Q parametrization, similar to the how Dueling Network (Wang et al., 2016; Tang et al., 2023; Han et al., 2024) parametrizes the scores for the next tokens given the prefix:

$$\hat{r}_{\text{RAD-Q}}(v|x) = \underbrace{\hat{r}_b(h(x))}_{\text{baseline}} + \Delta\hat{r}(e(v)|h(x)), \tag{8}$$

where the baseline predicts the score for the prefix $x$ and $\Delta\hat{r}$ predicts how observing a next token $v$ changes the score. Particularly, we use a *linear parametrization*:

$$\hat{r}_b(v|x) := \langle h(x), w \rangle, \qquad \Delta\hat{r}(e(v)|h(x)) := \langle h(x), We(v) \rangle. \tag{9}$$

Here, we introduced two attribute-specific parameters: $w \in \mathbb{R}^d$ for modeling the baseline reward score of the prefix, and $W \in \mathbb{R}^{d \times d}$ to model marginal rewards for each next token candidate.

**Rank bottleneck of RAD-Q.** Now it is clear that in contrast to RAD-V, RAD-Q (as defined in Equation (9)) performs a *low-rank matrix factorization* of $P_\Omega(R)$:

$$\hat{R}_{\text{RAD-Q}} = H(w\mathbf{1}^T + WE) = HA, \tag{10}$$

where we stack all context representations $x'$ into $H \in \mathbb{R}^{N \times d}$ and all next token representations into $WE \in \mathbb{R}^{d \times |V|}$, and $\mathbf{1}$ is a column $d$-vector of all ones. By the rank inequality,

$$\text{rank}(\hat{R}_{\text{RAD-Q}}) = \text{rank}(HA) \leq \min(\text{rank}(H), \text{rank}(A)) \leq d, \tag{11}$$

meaning that if $\min \text{rank}_\Omega(R) > d$, RAD-Q **cannot** possibly perfectly reconstruct $P_\Omega(R)$ no matter how flexible $h(x)$ is. In terms of $\varepsilon$-rank, if $\min \text{rank}_{\Omega,\varepsilon}(R) > d$, then RAD-Q cannot fit the data with training loss less than $\varepsilon$. In the language modeling literature, the rank bottleneck problem is known as the *softmax bottleneck* (Yang et al., 2018) and mitigation strategies are well-studied (Ganea et al., 2019; Chang & McCallum, 2022).

In the experiments (§4), we empirically demonstrate that distillation of RAD-V into low-rank RAD-Q can match the performance of the more flexible RAD-V on the two standard controlled generation benchmarks.

### 3.3 RAD-Q training

To train RAD-Q, we rely on the RAD approach to train a reward model. We split $x$ into a last token and remaining prefix: $x = [x', v]$. We pass $x'$ as input to the model, and $v$ indexes output embeddings (9). We consider two types of experiments: training RAD-Q on original responses from the dataset, and a distillation experiment, where we train RAD-Q to predict the scores of less efficient RAD.

For the first type of experiment, we train RAD-Q on the responses from the dataset using the weighted squared loss:

$$\mathcal{L}(\hat{r}(v|x'), y, \lambda) = \lambda(\hat{r}(v|x') - y)^2 \tag{12}$$

For the second type of experiment, we train an RAD-Q student to approximate the less efficient RAD-V teacher $\tilde{r}(x)$ (a frozen trained RAD) using the *distillation loss* (Hinton et al., 2015):

$$\mathcal{L}_{\text{dstl}}(\hat{r}(v|x'), \tilde{r}(x)) = (\hat{r}(v|x') - \tilde{r}(x))^2. \tag{13}$$

A reward model can only observe a limited number of next tokens $v$ given $x$ during finetuning. While the loss defined above provides a positive signal for some tokens $v$, it might be beneficial to regularize the prediction for other (unrelated) tokens, including rare or unseen tokens. In our parametrization (8), it is natural to push the predicted reward towards the baseline for unrelated tokens. We regularize the prediction of RAD-Q to be close on average to the prefix baseline by forcing $\Delta\hat{r}$ to be close to 0 for randomly sampled token candidates:

$$\mathcal{L}_{\text{reg}}(h(x)) = \mathbb{E}_{v' \sim \text{Uniform}[V]} \left[ \Delta\hat{r}(e(v')|h(x)) \right]^2, \tag{14}$$

where we use one sample of $v'$ for each prefix position, sampling uniformly from the vocabulary. Particularly, a regularized model can learn to *abstain* by predicting the baseline score for each next token candidate, which will not change the distribution of a base model.

## 4 Experiments

### 4.1 Controlled generation

We follow previous work (Deng & Raffel, 2023; Liu et al., 2021) and evaluate RAD-Q on two controlled generation tasks: detoxification and sentiment control.[1]

In our experiments, we guide the decoding from a base model using a smaller finetuned reward model with the same tokenizer. Namely, we guide GPT-2-Large using a reward model finetuned from GPT-2-Small, and we guide the LLaMa-2-(7b/13b) (Touvron et al., 2023) base language model with a reward model finetuned from TinyLLaMa (Zhang et al., 2024). We finetune all parameters of the reward models except input/output embeddings, which remain frozen (in hope of improving generalization to unseen tokens).

We conduct experiments in two regimes: first, by distilling less efficient RAD-V (Deng & Raffel, 2023) using $\mathcal{L}_{\text{dstl}}$ loss (13); second, by training a reward model from scratch on the responses from the datasets using cumulative loss $\mathcal{L}$ (12). In both settings, we use additional regularization $\mathcal{L}_{\text{reg}}$ by default. For evaluation, we perform guided decoding using *top-k* sampling from the categorical distribution defined in (2), where *top-k* candidates are selected taking $k$ largest logits of the base model at the current decoding step.

### 4.2 Detoxification

For the detoxification evaluation, we follow previous work (Deng & Raffel, 2023; Liu et al., 2021) and evaluate samples from guided decoding given a 10k subset (Liu et al., 2021) of prompts from the RealToxicityPrompts dataset (Gehman et al., 2020). We follow Deng & Raffel (2023) and Liu et al. (2021) and finetune our model on 2M pairs of text and continuous 'toxicity' responses between 0 and 1 from the Jigsaw Unintended Bias in Toxicity Classification challenge (cjadams et al., 2019). Like previous work, we train our model on 7 independent responses ('toxicity', 'severe toxicity', 'obscene', 'identity attack', 'insult', 'threat', 'sexual explicit') with different head parameters $w_i, W_i, i \in \{1, ..., 7\}$ for each sub-task. During decoding, we only use the 'toxicity' predictor. For the distillation experiment, we use the same dataset, and the released toxicity discriminator from Deng & Raffel (2023) as a teacher.

During decoding, we sample 25 continuations generating at most 20 new tokens. To evaluate toxicity, we use an external closed-source toxicity classifier *Perspective API* (Lees et al., 2022), and following previous work (Deng & Raffel, 2023; Liu et al., 2021), we rely on the *Maximal Average Toxicity* metric, which is the maximal toxicity score value over 25 samples for a given prompt, averaged over the set of 10k prompts. We also report *Toxic Rate*, which is calculated as the probability that at least one out of 25 continuations is toxic according to Perspective API (toxicity score $> 0.5$); and *Diversity* score, which is the average number of distinct $n$-grams normalized by the length of text (Li et al., 2018). To evaluate the fluency of model generations, we follow

---

[1]Our code is available at `https://github.com/serjtroshin/rad-q`

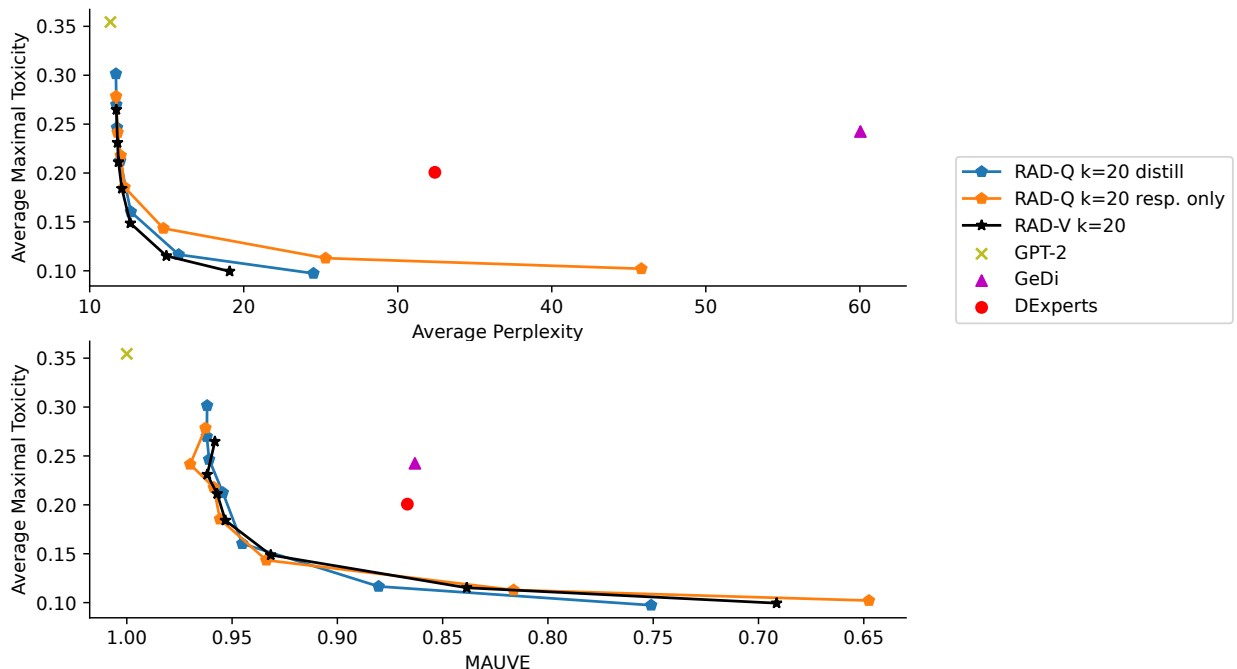

**Figure 4:** RAD-Q student (distill) shows comparable toxicity/fluency trade-off with the teacher RAD-V, where the RAD-Q student closely matches the performance of the teacher RAD. RAD-Q trained on original responses (RAD-Q resp. only) shows slightly worse fluency and similar toxicity level. We rerun the evaluation for RAD-V, GeDi and DExperts with an up-to-date Perspective API classifier. We include the results with other baselines from Deng & Raffel (2023) in Figure 14 (see Appendix G.1.1).

previous work (Liu et al., 2021; Deng & Raffel, 2023) and report the average perplexity of the GPT-2-XL when generating from the GPT-2-Large model; and the OLMo[2] to evaluate the LLaMa family as in Lovelace et al. (2024). As an additional fluency metric, we report MAUVE (Pillutla et al., 2021) to measure the distance between unguided and guided generations (details in Appendix F). In the experiments, we will look at the toxicity/fluency trade-off, alternating the weight $\beta$ of the discriminator (see Table 3 and Table 4). We expect to obtain a model with both low toxicity according to the Perspective API, and high fluency.

Since toxicity scores from the Perspective API can change overtime, which can complicate the evaluation, in Appendix G.3.1 we evaluate our detoxification models with an open-weight toxicity classifier,[3] where we observe the same relative results as with Perspective API scores.

## 4.3 Sentiment control

For sentiment control, we follow previous work (Li et al., 2018; Sudhakar et al., 2019; Liu et al., 2021; Deng & Raffel, 2023) to evaluate the samples given a prompt from one of the three categories: 2.5K *negative*, 5K *neutral*, and 2.5K *positive* prompts from OpenWebText (Gokaslan & Cohen, 2019). To finetune RAD-Q on responses only, we follow Deng & Raffel (2023) and finetune our model on millions of reviews from the Amazon Polarity (Zhang et al., 2015) and SST-2 (Socher et al., 2013) datasets. To distill the sentiment discriminator of Deng & Raffel (2023), we use text examples from the Amazon Polarity dataset. Additional training details are provided in Appendix E.

For evaluation, we follow Deng & Raffel (2023), and use the average *Positive Rate* metric *w.r.t.* the finetuned DistilBERT classifier (Sanh et al., 2019) provided via HuggingFace.[4] As in the toxicity task, we use GPT-2-

---

[2]https://huggingface.co/allenai/OLMo-1B

[3]https://huggingface.co/nicholasKluge/ToxicityModel

[4]https://shorturl.at/9MqDp

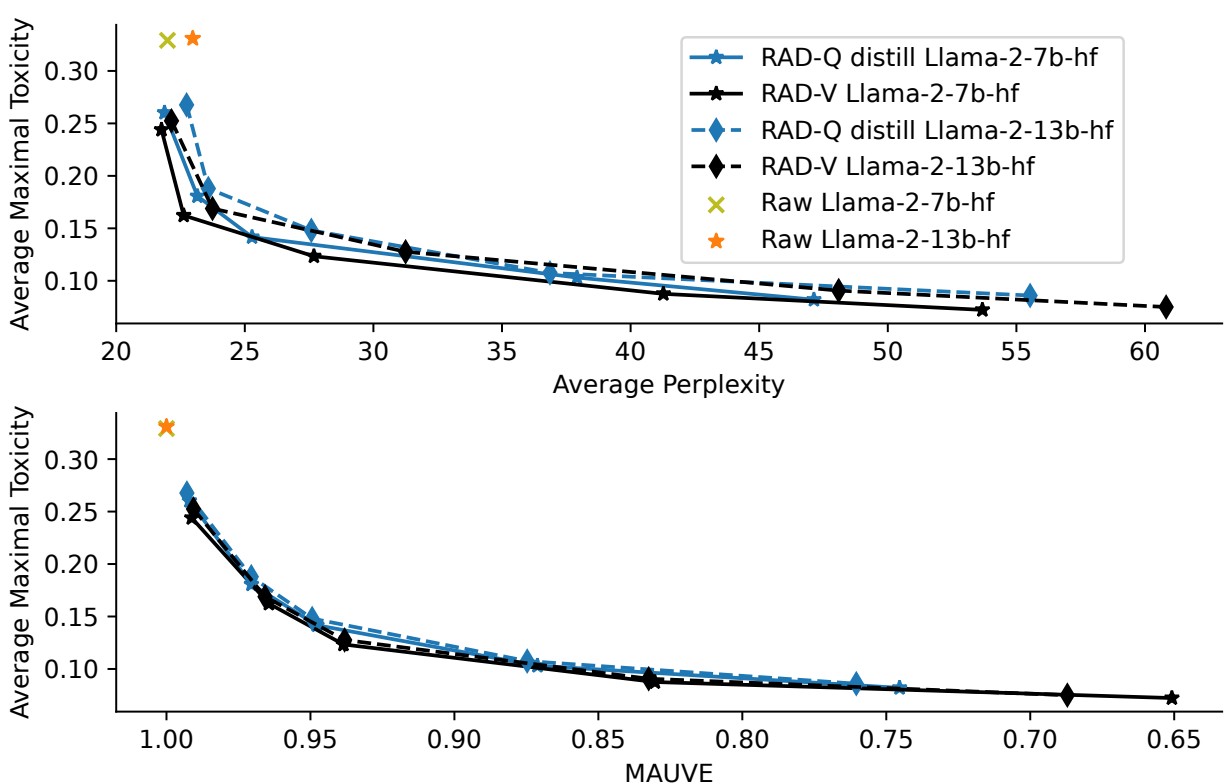

**Figure 5:** Detoxification results with Perspective API toxicity classifier and LLaMa-2 model. RAD-V and RAD-Q (distill) demonstrate similar performance $w.r.t.$ two fluency metrics: average perplexity and MAUVE. Performance remains consistent across different models sizes of base LLaMa-2 model.

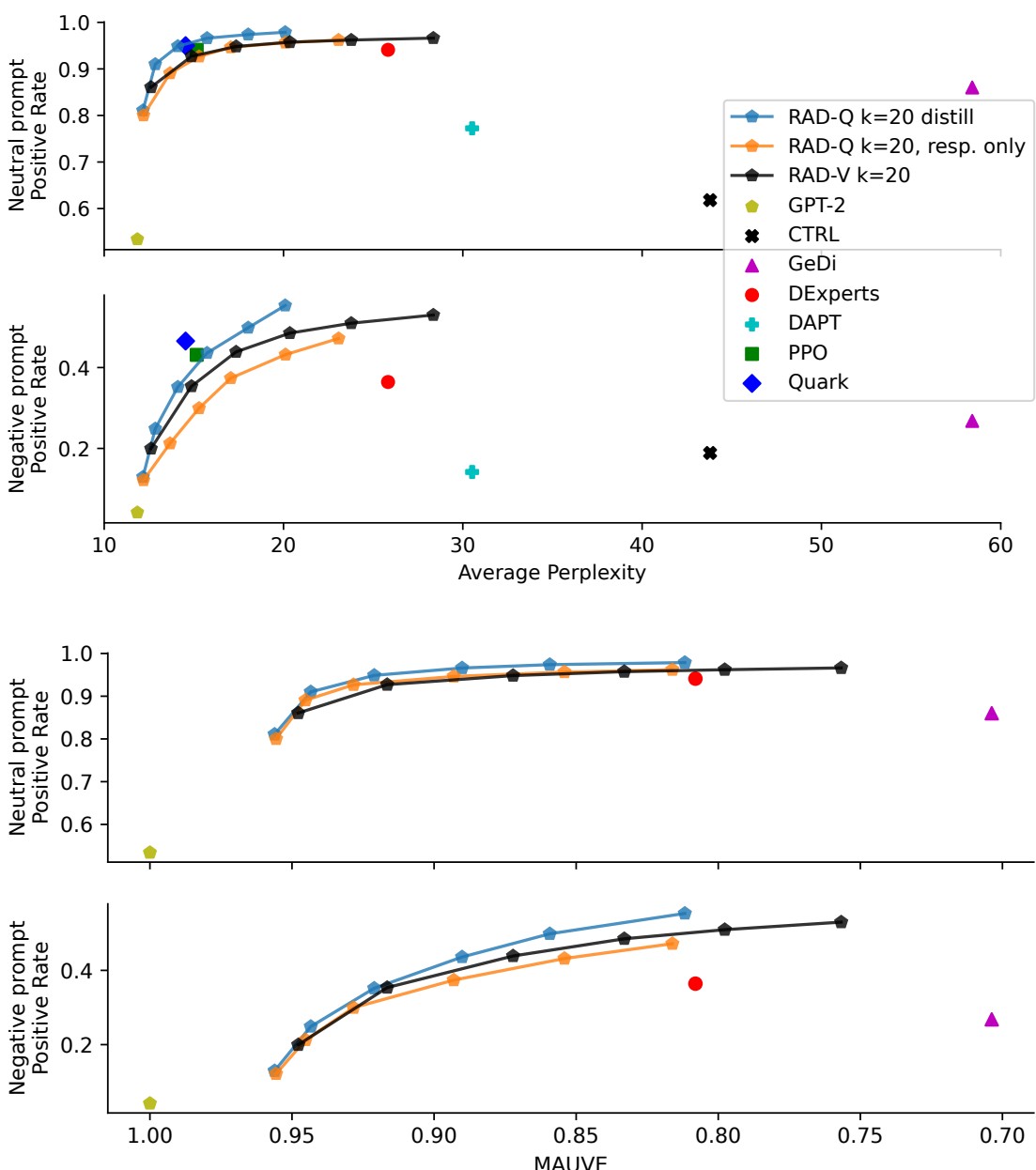

**Figure 6:** For the sentiment control task, RAD-Q trained on responses only lags slightly behind the RAD-V baseline, while student RAD-Q outperforms the teacher RAD-V model. For the plot with average perplexity, we include the results from Deng & Raffel (2023) for other baselines for reference.

XL/OLMo and MAUVE to evaluate the fluency of the sampled continuations, and we expect to obtain a high Positive Rate and high fluency.

### 4.4 Results

To compare RAD-V and RAD-Q, we rely on the methodology of Deng & Raffel (2023) and Liu et al. (2021), and visualize the trade-off plots for both models varying the control parameter $\beta$. Namely, each point in the figure will represent two metrics: toxicity/sentiment along the vertical axis and fluency along the horizontal

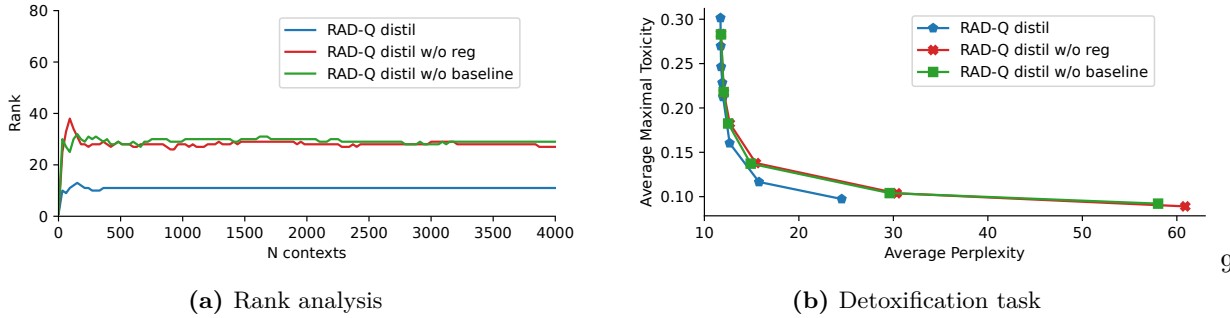

**(a)** Rank analysis                    **(b)** Detoxification task

**Figure 7:** Ablation experiment for distilled RAD-Q, on the detoxification task with top-$k$=20. On the right, we observe that regularization towards the baseline results in better fluency of generated samples. On the left, we observe that regularization lowers the rank of the model's outputs rank($\hat{R}_{\text{RAD-Q}}$).

axis. From this plot, we can read, *e.g.*, what fluency (perplexity/MAUVE) can be achieved for a given 'target' toxicity. To compare two models, we compare their curves (in the same plot). Our hypothesis is that RAD-Q will perform similar to the more flexible RAD-V approach, meaning that the trade-off plots for these models will be close to each other.

**Detoxification.**   For the detoxification task (Figure 4 with GPT-2, Figure 5 with LLaMa-2), our efficient student (RAD-Q) closely follows the RAD-V teacher for toxicity control/fluency trade-off. We observe that RAD-Q trained on responses only shows slightly worse fluency *w.r.t.* average perplexity for lower levels of toxicity. For completeness, in Figure 14, we include the results for other baselines from Deng & Raffel (2023) computed for an older version of Perspective API. For guided decoding from the LLaMa-2-(7b/13b), we observe that again RAD-Q closely follows RAD-V in terms of toxicity/fluency trade-off (see Figure 5 in Appendix G.1.1).

**Sentiment control.**   From the results on the sentiment control task in Figure 6, we observe that the RAD-Q student model shows slightly better trade-off than the RAD-V teacher model, closely following approaches that require training using feedback from the evaluation pipeline (Lu et al., 2022, Quark), (Stiennon et al., 2020, PPO). Again, RAD-Q trained on original responses slightly lags behind but still performs competitively compared to other guided decoding baselines.

**Summary.**   First, we observe that distilled RAD-Q can match or even outperform RAD-V, which confirms that *Q*-style parametrization is expressive enough for our controlled generation scenarios. Second, distilling the RAD-V teacher into the RAD-Q student results in slightly higher quality compared to training RAD-Q on original responses. One difference is that when training from data, we will see short contexts multiple times with different reward responses and must implicitly converge to their average, while in distillation, the teacher already performs this compression and provides a single deterministic target $\hat{r}(v|x)$ for every context $(x, v)$. We conjecture that this may lead to better-trained distilled models.

### 4.5   Ablation

In this section, we investigate the effect of adding the baseline component Equation (9) and of regularization Equation (14). In Figure 7, we experiment with the distilled version of RAD-Q and observe that turning off regularization, or further removing the baseline from the parametrization results in still adequate but slightly worse fluency as measured by perplexity, and a comparable toxicity decrease. By further analyzing the ranks of $R_{\text{RAD-Q}}$ with and without regularization, we observe that regularization effectively decreases the rank of $R_{\text{RAD-Q}}$, which might explain the higher fluency of regularized models. Particularly, strong regularization would result in the model always predicting the baseline score for each of the next tokens (corresponding to the rank-1 output), which does not modify the original distribution of the model (the best fluency).

| Model | N calls |
|-------|---------|
| GeDi (Krause et al., 2021) | 1 |
| DExperts (Liu et al., 2021) | 2 |
| RAD-V (Deng & Raffel, 2023) | $k$ |
| RAD-Q (Ours) | 1 |

**Table 1:** Number of input tokens a discriminator model needs to process for a single decoding step with $k$ next token candidates. All included models are based on the unidirectional Transformer (Vaswani et al., 2017) and support the caching of prefix activations.

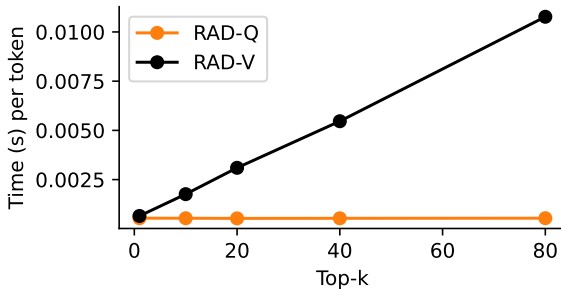

**Figure 8:** RAD-V processes the $k$ next token candidates separately as input requiring more time compared to RAD-Q, which relies on the output layer to obtain the scores for all next tokens.

### 4.6 Efficiency

We consider using a reward model to compute the scores for $k$ candidate tokens at each of $L$ steps of decoding. Similar to RAD-V (Deng & Raffel, 2023), RAD-Q is based on the *unidirectional* Transformer architecture (Vaswani et al., 2017), which means that we can cache the prefix activations during decoding. To compute the prediction for $k$ next token candidates $v$ given a prefix $x$, RAD-V needs to pass $k$ next tokens as *input* to the Transformer model, thus RAD-V processes $O(Lk)$ tokens during decoding. In contrast, RAD-Q only processes $O(L)$ tokens as input to the Transformer model and relies on the output layer to efficiently compute the scores for all next token candidates. In Table 1, we summarize how many tokens external expert models process during top-$k$ decoding. In Figure 8, we measure the time per generated token when running the decoding for the toxicity task with RAD-Q and RAD-V (Deng & Raffel, 2023) on a single RTX A6000 GPU.

## 5 Related Work

There are multiple approaches that investigate how to finetune a language model using attribute-conditioned data (desired/undesired examples). Keskar et al. (2019) finetunes a language model using control prompts. More recent approaches (Schulman et al., 2017; Stiennon et al., 2020; Lu et al., 2022) perform finetuning while regularizing the weights of the model to stay close to pretrained weights. Despite the efficiency of decoding, these methods might require more resources for finetuning if the language model is large, or might even be unusable if we only have access to the top-$k$ logits of base language model via an API. Unlike finetuning, alternative approaches keep the language model untouched and use external models to guide the decoding from the base language model. Dathathri et al. (2019) use the gradients from a discriminator to modify the prefix activations of the base model during decoding. However, gradient-based methods are costly to use during decoding since they require backpropagating through the large base model.

Closest to our work are *gradient-free* guided decoding methods, where we have access to the frozen base language model and use external models to guide the sampling process from the base model. Among Q-parametrized models, GeDi (Krause et al., 2021) uses class-conditioned language models as discriminators to augment the decoding and efficiently compute the scores for next token candidates. DExperts (Liu et al., 2021) improves the quality of GeDi introducing an ensemble of two class-conditioned language models finetuned on desired and undesired data. Cao et al. (2022) finetune a *Q*-style model to reduce the probability of reaching undesired terminal states. In concurrent work, Xu et al. (2025) propose a *Q*-style model, where they show that Q and V style models are equivalent, assuming the logits come from a universal approximator. While true, this analysis omits the rank bottleneck present in practical models, which we analyze in this work. We hope our work will inform this line of work of potential limitations of *Q*-style parameterized models.

*V*-style models are often used to re-rank the intermediate hypotheses *e.g.* for best-of-n sampling, which however requires a large pool of *completed* hypotheses (Sun et al., 2024). Deng & Raffel (2023), Sitdikov et al. (2022) argue to use discriminator models akin to Yang & Klein (2021) to guide the intermediate outputs

*during* the decoding, whereas Deng & Raffel (2023) propose an effective *V*-style parametrized reward model trained from labeled data examples. Sitdikov et al. (2022); Dekoninck et al. (2023) use available *bidirectional* Transformers to guide the base language model, which, however, requires recomputing all prefix tokens at each decoding step. To tackle this issue, RAD-V (Deng & Raffel, 2023) proposes a *unidirectional* model suitable for caching of prefix activations. Among RL-based approaches, Mudgal et al. (2024), Chakraborty et al. (2024) use a *V*-style model although at the same time they rely on the losses designed for a *Q*-style model (e.g. CD-Q from Mudgal et al. (2024)). To the best of our knowledge, there is little attention to the implied efficiency-quality trade-off that we study in our work. The closest to our analysis, Han et al. (2024) compare both parametrizations in relation to language tasks, where they empirically observe that *V*-style parametrization outperforms *Q*-style parametrization.

To summarize, we complement the previous work, by zooming in into the parametrization of an autoregressive reward model. We highlight the trade-off between efficiency and expressiveness of a reward model, and showcase that 1) in theory there is a gap in expressivity between *Q*-style and *V*-style models due to rank bottleneck, 2) for the tasks and datasets we consider, higher rank-expressiveness can be traded for higher efficiency without quality drop when using distillation. We hope that our analysis will inform future work on the design choices of autoregressive reward models.

## 6 Conclusion

We review the recently proposed RAD-V approach of training a reward model for the guided decoding, and we reformulate it as the incomplete reward matrix learning problem. In the light of the rank analysis of the reward matrix, we observe that the high flexibility of RAD-V might not overweight its lower efficiency during decoding. We revisit the low-rank parametrization style of reward models in application to RAD, and demonstrate the effectiveness of a more efficient low-rank RAD-Q parametrization. We thus bridge the gap between two paradigms of training external expert models, demonstrating that we can have both efficient and effective controlled generation. At the same time, we precaution from indiscriminately choosing low-rank parametrization by highlighting the cases when the incomplete reward matrix has higher minimal rank.

## Acknowledgments

This publication is part of the project VI.Veni.212.228 of the research program 'Veni', which is financed by the Dutch Research Council (NWO); and is part of 'Hybrid Intelligence: augmenting human intellect' (https://hybrid-intelligence-centre.nl) with project number 024.004.022 of the research program 'Gravitation' which is (partly) financed by the Dutch Research Council (NWO).

We thank Wilker Aziz, Bryan Eikema, Caio Corro, and members of LTL and CLTL for fruitful discussions and feedback. We also thank the Perspective API team for increasing the API quota for us.

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

# Appendix

Warning: (the last page of) this appendix contains model-generated text conditioned on high toxicity contexts.

## A   Limitations

The models discussed in this work can only reduce the probability of generating the toxic responses, not prevent it. Moreover, evaluation of toxicity is far from perfect, and even a very low toxicity score from automatic evaluation such as Perspective API does not necessary mean that the sample is 'safe'. Furthermore, we should not exclusively rely on toxicity when evaluating the safety of samples from language models due to the complexity and variability of language. It is also not clear that by reducing toxicity, we are not introducing other harms. Furthermore, both RAD-V and our models represent low-rank $\hat{R}$ and further qualitative research is needed to investigate whether certain toxicity patterns require high rank to represent them.

## B   Reward Matrix

To train a reward model, we use weighted mean squared loss, for which the weighted mean recovers the minimum:

$$r^* = \arg\min_r \sum_{\lambda,y} \lambda(r - y)^2 = \frac{\sum_{\lambda,y} \lambda y}{\sum_{\lambda,y} \lambda} \tag{15}$$

*Proof.* $\frac{\partial}{\partial r} \sum_{\lambda,y} \lambda(r-y)^2 = \sum_{\lambda,y} \frac{\partial}{\partial r}[\lambda(r-y)^2] = \sum_{\lambda,y} 2\lambda(r-y) = 2(r\sum_{\lambda,y}[\lambda] - \sum_{\lambda,y}[\lambda y]) = 0$. Hence, $r^* = \frac{\sum_{\lambda,y} \lambda y}{\sum_{\lambda,y} \lambda}$ ◻

## C   Factorization of $P_\Omega(R)$

Any matrix $R \in \mathbb{R}^{N \times |V|}$ can be factored as $R = UV^T$ with $U, V$ of dimensions $N \times q; |V| \times q$. If $R$ is *incomplete*, then there are in general multiple possible factorizations of $P_\Omega(R)$ compatible with the observed values.

### C.1   Rank-1 case

To get a better intuition why the incompleteness of $P_\Omega(R)$ allows us to find a compatible factorization with lower minimal rank, consider a simple example. If we only know 1 element per row of $R$, then the minimal rank of $P_\Omega(R)$ is equal to 1. To prove this, consider completing $P_\Omega(R)$ such that each row is filled with the same element (the only one known for this row):

$$\begin{pmatrix} 1 & ? & ? \\ ? & 4 & ? \\ ? & ? & 3 \end{pmatrix} \rightarrow \begin{pmatrix} 1 & 1 & 1 \\ 4 & 4 & 4 \\ 3 & 3 & 3 \end{pmatrix}$$

### C.2   A case of a single missing value.

Here we prove that introducing a missing value in a random $k$-by-$k$ matrix will almost always (with probability 1) allow a completion of rank at most $k - 1$.

*Proof.* Let $R$ be a $k$-by-$k$ matrix, and without loss of generality, by permuting rows and columns we may assume the unknown element is at position $(1, 1)$. Consider completing it with a value $x$, and denote the

completed matrix as $A$. Let $A_{ij}$ be the $ij$ minor of $A$, *i.e.* the determinant of the submatrix left after removing row $i$ and column $j$.

We can expand $\det(A) = xA_{11} + a_{21}A_{21} + \ldots + a_{k1}A_{k1}$. In order for $A$ to not be full rank, there needs to be a solution in $x$ to $\det(A) = 0$. If $A_{11}$ is not $0$, then $x = -(a_{21}A_{21} + \ldots + a_{k1}A_{k1})/A_{11}$ will lead to a completion of rank less than $k$ regardless of any of the specific values. However, if $A_{11}$ is $0$ but $a_{21}A_{21} + \ldots + a_{k1}A_{k1}$ is not $0$, then there is no solution for $x$ that would result in a determinant of $0$ and thus the rank of $A$ must be $k$. The set of matrices $R$ satisfying $R_{11} = 0$ has zero Lebesgue measure inside $\mathbb{R}^{k \times k}$, and thus for any distribution over matrices with full support, sampling such $R$ is a zero-probability random event. □

## C.3 Estimating the minimal rank of the data

Empirically calculating minimal rank is challenging due to the very large number of prefixes (row of the matrix), particularly, a large portion of the prefixes have unique continuations. We show how we simplify the minimal rank estimation by considering only the prefixes with two or more continuations, and demonstrate that partially observed $\hat{R}$ can be fit with a low rank matrix factorization.

Given a training dataset of responses and text utterances, there will be many unique prefixes, for which the Example 1 is applicable. We can therefore reduce the complexity of rank estimation by skipping these prefixes, as shown in the next result.

**Lemma 3.** Let $(R, \Omega)$ be a partially-observed matrix and $\Omega_2 \subseteq \Omega$ denote the subset of observed indices that have at least one other index in the same row. Then,

$$\min \operatorname{rank}_{\Omega,\varepsilon}(R) \leq 1 + \min \operatorname{rank}_{\Omega_2,\varepsilon}(R) \tag{16}$$

*Proof.* Permute rows such that all rows with a single index are grouped together. We have a block-incomplete matrix where the top block admits a completion of rank 1, and the bottom block admits a completion of rank $\min \operatorname{rank}_{\Omega_2,\varepsilon}(R)$. The rank of the stacked completions is no more than the sum of the ranks of the completions.

This result reduces the problem of estimating $\min \operatorname{rank}_{\Omega,\varepsilon}(R)$ to the possibly smaller problem of estimating $\min \operatorname{rank}_{\Omega_2,\varepsilon}(R)$ (since any fully-unobserved rows and columns can be skipped.)

We now numerically verify that there exist low-rank factorizations compatible with $P_{\Omega_2}(R)$ within $\varepsilon$. Finding such a factorization of rank $r - 1$ implies, by the previous lemma, that $R$ is of minimal rank $r$ *w.r.t.* $\Omega$ and thus that the training dataset $D_f$ can be fit by a reward model with rank bound by $r$, regardless of the specifics of said model. In general, finding minimal rank factorization of incomplete matrices is known to be NP-hard, and usually convex relaxation such as minimization of the nuclear norm is considered (see Nan (2009)). To factorize a partially-observed matrix, we use the *soft impute* alternating least squares algorithm (Mazumder et al., 2010; Hastie et al., 2015).[5] Given a matrix $X \in \mathbb{R}^{n \times m}$ with observed indices $\Omega$, this algorithm solves

$$\text{minimize } \left\| P_\Omega(X - AB^\top) \right\|_F^2 + \lambda(\|A\|_F^2 + \|B\|_F^2)$$
$$\text{with respect to } A \in \mathbb{R}^{n \times k}, B \in \mathbb{R}^{m \times k} \tag{17}$$

by alternating between efficient soft SVD updates for $A$ given $B$ and $B$ given $A$. We optimize for 1000 iterations with a trace norm penalty of $\lambda = 10^{-4}$. This penalty induces a small bias but improves convergence. At convergence, $A$ and $B$ constitute a *certificate* of the minimal numerical rank of $X$ *w.r.t.* $\Omega$.

Results reported in table 2 provide strong evidence that both datasets can be approximated well by low rank matrices, close to the $10^{-6}$ resolution of 32-bit floating point numbers. The reported MSE values for rank $d - 1$ can be interpreted as reachable lower bounds of the MSE training loss of a RAD-Q transformer with hidden dimension $d$ on the respective training data.

---

[5] https://cran.r-project.org/web/packages/softImpute/index.html

| rank | detox. | sentiment | helpfulness | safety |
|------|--------|-----------|-------------|--------|
| 0 | 4.6e-2 | 4.9e-1 | 8.4 | 0.42 |
| 255 | 3.4e-5 | 3.4e-4 | 1.04e-08 | 5.08e-08 |
| 511 | 7.7e-6 | 9.7e-5 | 1.45e-09 | 5.77e-10 |
| 767 | 4.7e-7 | 6.6e-7 | 1.28e-09 | 4.81e-10 |

**Table 2:** Mean squared errors of low-rank matrix completion of the Jigsaw (detox) and Amazon review polarity (sentiment) datasets following the methodology described in appendix C.3. Additionally, we report MSE for HelpSteer (helpfulness) dataset (Wang et al., 2024b) and BeaverTails (safety) dataset (Ji et al., 2023), which are commonly used for reward model training (Wang et al., 2024a). The zero rank row corresponds to predicting the zero matrix. All datasets can be approximated well by low rank models. For the ranks, we use multiples of $256 - 1$, because one rank is reserved to handle the single-occurrence contexts.

# D  Rank expressivity of $\hat{R}_{\text{RAD-V}}$ and $\hat{R}_{\text{RAD-Q}}$

## D.1  Rank expressivity of RAD-$V$

In this experiment, we empirically verify that RAD-V is capable to approximate $P_\Omega(R)$ matrix with $\text{rank}(P_\Omega(R)) > d$, where $d$ is the dimensionality of the model. We finetune RAD-V initialized from the GPT-2-Small (with $d = 764$) on a synthetic data constructed as in example 3. We generate $P_\Omega(R), n = 1024 > d$, an incomplete matrix of size 1024 with unknown elements above the diagonal, ones on the diagonal, and zeros below the diagonal. With this construction, $\min \text{rank}_\Omega(R) = n$ and thus greater than the model dimension.

We verify that we can train RAD-V to fit this train matrix obtaining the training loss (MSE) less than $5 \cdot 10^{-6}$.

## D.2  Rank expressivity of RAD-$Q$

RAD-Q approximates $P_\Omega(R)$ as a product of two rank $d$ matrices, hence RAD-Q cannot reconstruct the data perfectly, which has higher rank; RAD-Q obtains MSE $> 0.0001$.

We thus conclude that RAD-V (in contrast to RAD-Q) is indeed capable of representing $P_\Omega(R)$ matrices with a rank higher than $d$.

## D.3  Real data experiments

For the experiment with the real datasets for the detoxification and sentiment control tasks, in Figure 9, we numerically measure the rank of $R_{RAD-V}$ and $R_{RAD-Q}$, and observe that both RAD-Q and RAD-V learn low-rank reward matrices. We thus conclude that both these models have needed capacity to represent the incomplete $P_\Omega(R)$ matrices obtained from the datasets.

## D.4  Numerical rank

To compute rank of $n \times m$ matrix, we use the default cutoff in Numpy[6] and PyTorch[7] at the time of writing, which is to say we count only singular values above $\max(m, n)\varepsilon\sigma_1$, where $\varepsilon$ is the machine epsilon for the corresponding data type, i.e., the difference between 1.0 and the next smallest representable number larger than 1.0, and $\sigma_1$ is the largest singular value.

There are potential issues that may arise when computing the numerical rank. One issue is that the singular values, especially for the matrices coming from 32bit float precision neural network, will not be exactly zero, so this is why libraries like Numpy or PyTorch use a precision-based cutoff for singular values that should be considered indistinguishable from zero; we use the default such parameters. The other issue is that the

---

[6] https://numpy.org/doc/stable/reference/generated/numpy.linalg.matrix_rank.html
[7] https://pytorch.org/docs/stable/generated/torch.linalg.matrix_rank.html

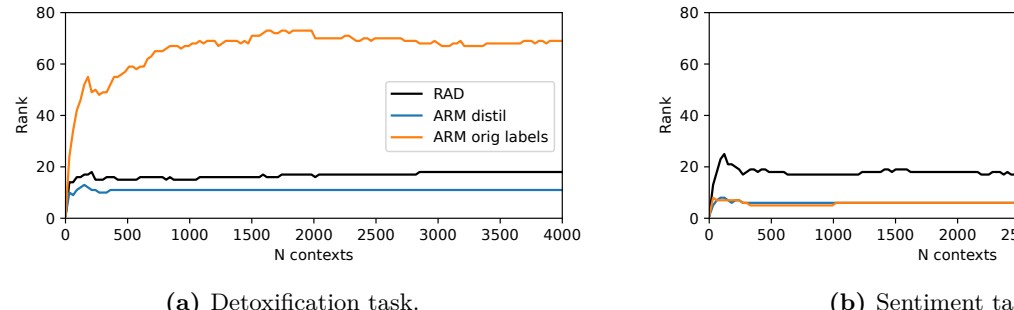

**(a)** Detoxification task.

**(b)** Sentiment task.

**Figure 9:** We numerically estimate the ranks of both $\hat{R}_{\text{RAD-V}}$ and $\hat{R}_{\text{RAD-Q}}$ increasing the number of training prefixes (rows of $\hat{R}$). In all cases, the ranks tend to be less than the model dimension $d = 764$. This means that rank-capacity of RAD-Q is sufficient to capture the training datasets for the detoxification and sentiment tasks.

number of rows in the reward matrices is very high and we follow the work of Finlayson et al. (2024) and estimate rank by sampling rows from the matrix. Different submatrices can have different ranks, but we sample i.i.d. to prevent this.

# E    Training Details

To train reward models, we reuse the hyperparameters from Deng & Raffel (2023), where possible. We finetune the reward models with Adam optimizer (Kingma & Ba, 2015) with $\beta_1 = 0.9, \beta_2 = 0.95, \epsilon = 1\text{e}{-}12$. We use weight decay 0.01, batch size 100, and the learning rate changes linearly from the initial value ($10^{-5}$ by default) to zero.

To train RAD-Q, we initialize the parameters with the pretrained GPT-2-Small/TinyLLaMa[8] weights, and freeze the shared input-output embedding parameters. Alternative strategy would be to use parameter efficient finetuning (Hu et al., 2022; Sidahmed et al., 2024).

## E.1    Detoxification

For the detoxification task, we finetune RAD-Q with the learning rate $10^{-5}$ for 5 epochs.

For the LLaMa-2, we additionally finetune RAD-V with the TinyLLaMa backbone for the fair comparison with RAD-Q.

## E.2    Sentiment Control

To finetune RAD-Q on responses only for sentiment control task, we first finetune the model with the learning rate $10^{-5}$ on the Amazon Polarity dataset, and then finetune it for 5 epochs on the SST-2 dataset with the learning rate $2\text{e}{-}6$. For distillation experiment, we finetune RAD-Q for 5 epochs with the learning rate $10^{-5}$ on Amazon Polarity dataset.

# F    MAUVE

To complement perplexity as a measure of fluency, we use MAUVE (Pillutla et al., 2021) as one of the fluency metrics. For reference texts, we take the generations of unguided model (GPT-2, or LLaMa-2-(7b/13b)). Thus, this metric should capture how close the distribution of the continuations of a guided model is to the distribution of the original language model. To calculate MAUVE, we follow recommendations of He et al. (2023) and use ELECTRA-large model to obtain the text representations. We use the hyperparameters of Pillutla et al. (2021): $c = 5$ for the scaling constant; $k$−means for the quantization algorithm with 500

---

[8]https://huggingface.co/TinyLlama/TinyLlama-1.1B-intermediate-step-1431k-3T

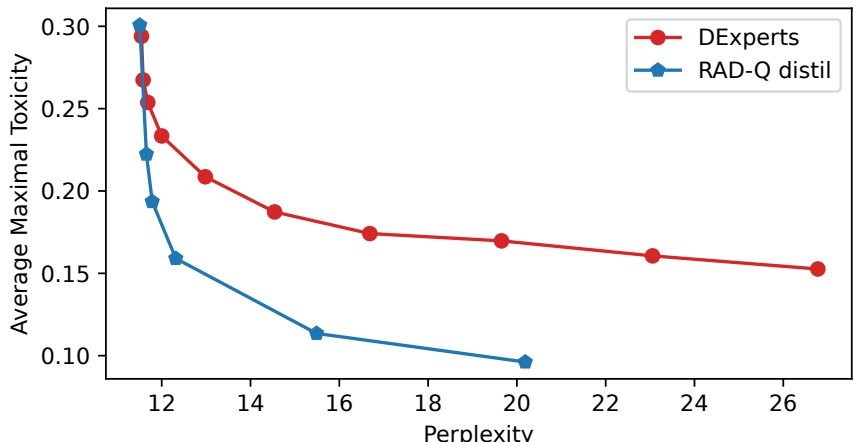

**Figure 10:** Comparison of toxicity/fluency trade-off between RAD-Q (distill) and DExperts. We rerun the sampling from these two models using *top-k* decoding with $k = 20$. Results are calculated over randomly selected 1000 prompts. We observe, that RAD-Q show better constraint satisfaction/fluency than DExperts.

iterations, and $n/10$ clusters where $n$ is the number of generations. To compute MAUVE, we use 1000 prompts from the evaluation dataset.

## G  Results

### G.1  Detoxification

#### G.1.1  Results with Perspective API classifier

In this section, we report full results with the Perspective API as a toxicity classifier.

**GPT-2.**  Results for the detoxification task with the GPT-2-Large base model and GPT-2-small reward model, are presented in Table 3.

We present the results for RAD-Q and RAD-V with *top-k* decoding with $k = 40$ in Figure 13. We observe similar relative performance of RAD-Q compared to RAD-V as in the experiment with $k = 20$, presented in the Figure 4.

**LLaMa-2.**  Results for detoxification task with LLaMa-2-(7b/13b) base model and TinyLLaMa reward model are presented in Figure 5 and Table 5.

**Baselines.**  Additionally, in Figure 14, we include results from Deng & Raffel (2023) for other baseline models (for an older version of Perspective API).

To highlight the difference between the RAD-Q and DExperts, we show the trade-off plot for DExperts model in Figure 10, varying the $\alpha$ scalar parameter for DExperts. As we can observe, the RAD-Q has better constraint satisfaction / fluency trade-off than DExperts model. We attribute this to the difference in the training objectives of the expert models (reward modeling or language modeling), as argued in (Deng & Raffel, 2023).

### G.2  Additional Ablation Results

#### G.2.1  Loss choice

We perform an ablation study for the choice of the loss function used to train RAD-Q Equation (12). There, we follow the approach of (Deng & Raffel, 2023), where they introduce the squared loss (see section 2.1

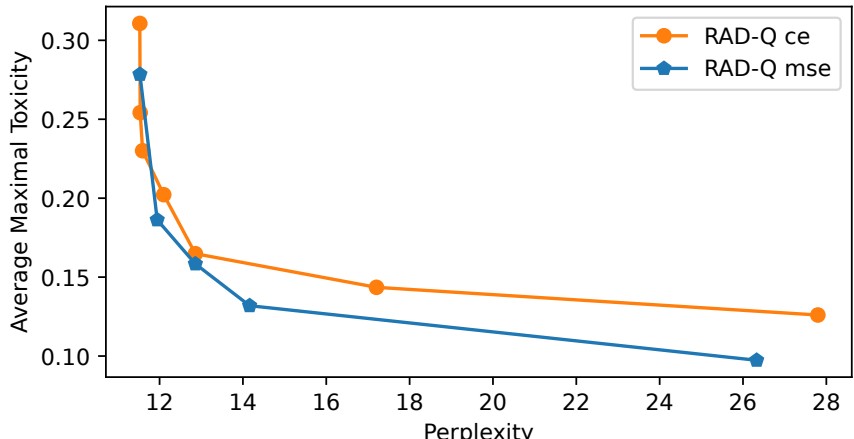

**Figure 11:** Comparison of RAD-Q model trained on original responses with squared loss vs with cross-entropy loss. We rerun the sampling from these two models using *top-k* decoding with $k = 20$. Results are calculated over randomly selected 1000 prompts. We observe, that RAD-Q trained with the squared loss show slightly better constraint satisfaction/fluency than RAD-Q trained with cross-entropy loss.

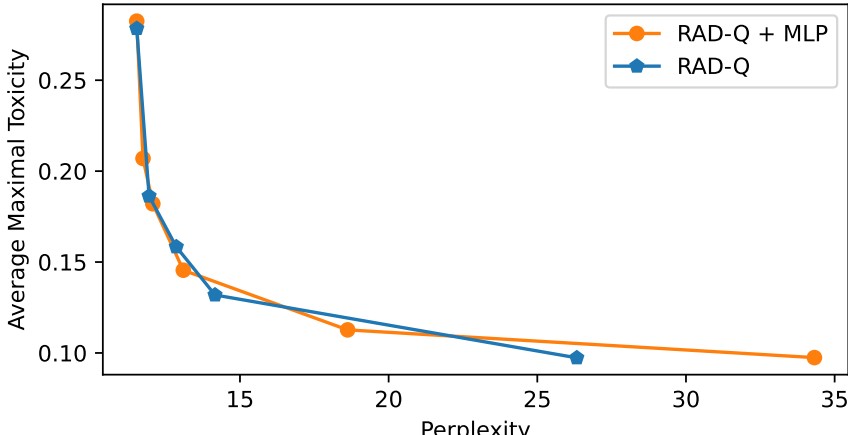

**Figure 12:** Comparison of RAD-Q trained on original responses with linear parametrization vs with non-linear MLP parametrization. We rerun the sampling from these two models using *top-k* decoding with $k = 20$. Results are calculated over randomly selected 1000 prompts. We observe, that both parametrizations perform very closely.

Unidirectional Reward Model). An alternative strategy would be to use the binary cross-entropy loss, using the fact that for our datasets responses $y$ are from $[0; 1]$ range:

$$\mathcal{L}_{ce}(\hat{r}(v|x'), y, \lambda) = \lambda(y \log \sigma(\hat{r}(v|x')) + (1 - y) \log(1 - \sigma(\hat{r}(v|x')))), \tag{18}$$

where we introduced $\sigma(x) = 1/(1 + e^{-x})$ function to softly map the predictions of RAD-Q into $[0; 1]$ range, which is also used during generation. Figure 11 demonstrates that the RAD-Q trained with the squared loss slightly outperform the RAD-Q trained with the binary cross-entropy loss.

### G.2.2   MLP vs Linear Parametrization

In this ablation, we consider replacing the linear parametrization of RAD-Q Equation (10) with a non-linear MLP parametrization:

$$\Delta \hat{r}_{\text{RAD-Q+MLP}}(x) := W_1 \sigma(W_2 E^T W^T h(x)^T), \tag{19}$$

where $W_1 \in \mathbb{R}^{d \times |V|}; W_2 \in \mathbb{R}^{|V| \times d}$. As we observe in Figure 12, MLP parametrization performs on par with the linear parametrization. We thus recommend using a more simple linear parametrization.

### G.3   Sentiment control

Here, in Figure 15, we include the additional results for the RAD-V and RAD-Q with *top-k* decoding with $k = 40$.

### G.3.1   Results with RoBERTa classifier

In addition to toxicity scores with Perspective API, we provide the results with the open-weight RoBERTa toxicity classifier (Corrêa, 2023) for the guided generation with GPT-2 (Figure 16 and Table 6) and the LLaMa-2 (Figure 17 and Table 7). We notice that results for the average maximal toxicity with RoBERTa are relatively similar to the result with Perspective API. We hope that with an open-weight classifier it will be easier for the community to directly compare to the published results without the need to recompute the API scores.

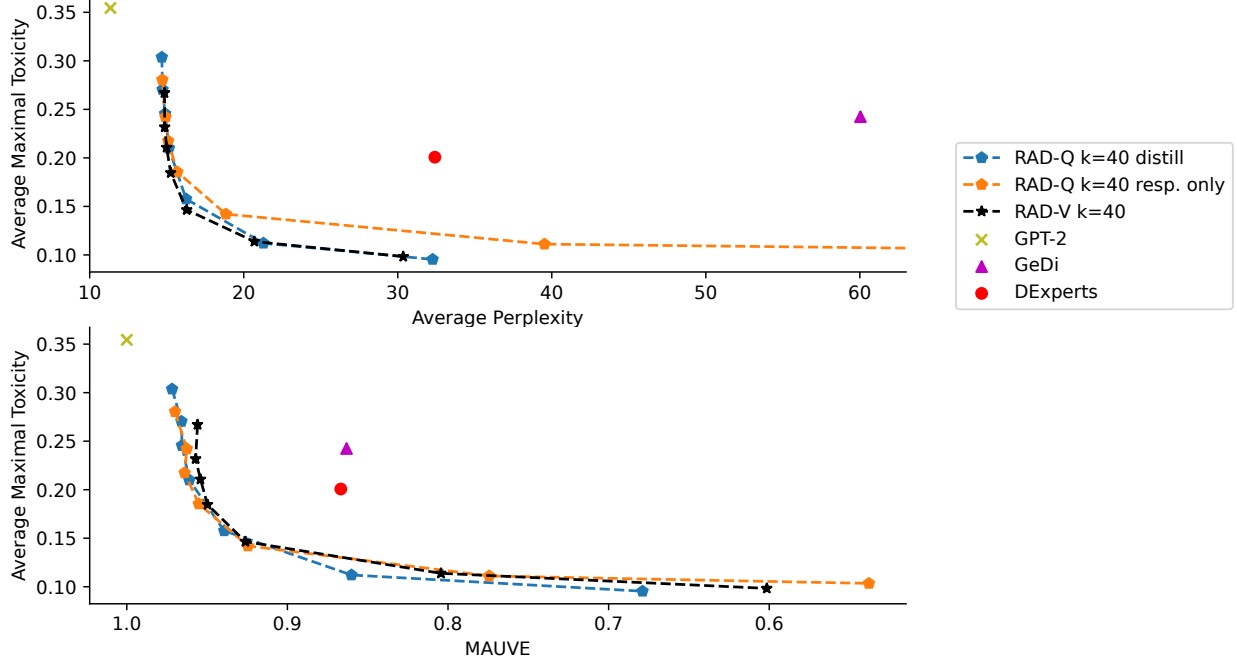

**Figure 13:** Additional results for detoxification task for RAD-Q and RAD-V with $k = 40$.

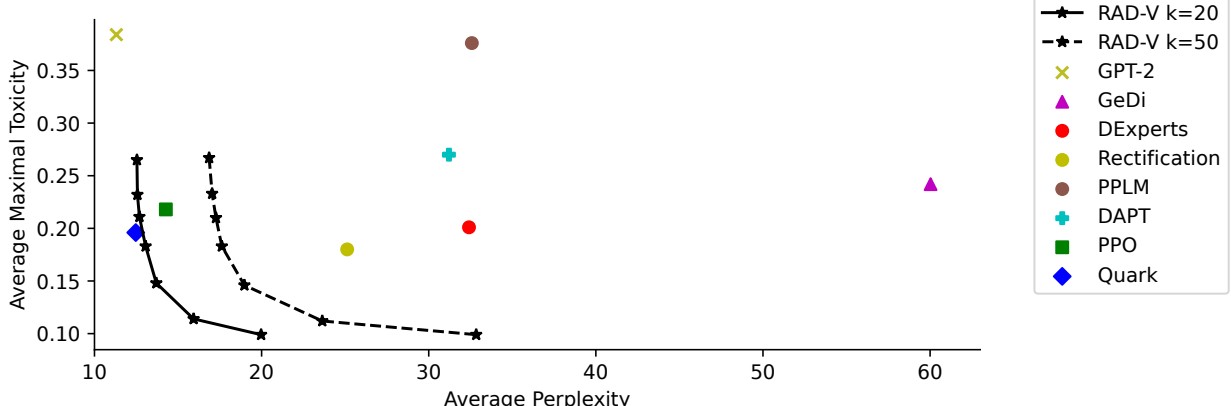

**Figure 14:** Detoxification results reported in Deng & Raffel (2023) with Perspective API with GPT-2-Large model (API queries made between May and June 2023).

### G.4   Sentiment Control

Results for sentiment control task with the GPT-2-Large are presented in Table 4.

### G.5   Generated Examples

Examples for the detoxification and sentiment control are presented in the Table 8, Table 9 and Table 10.

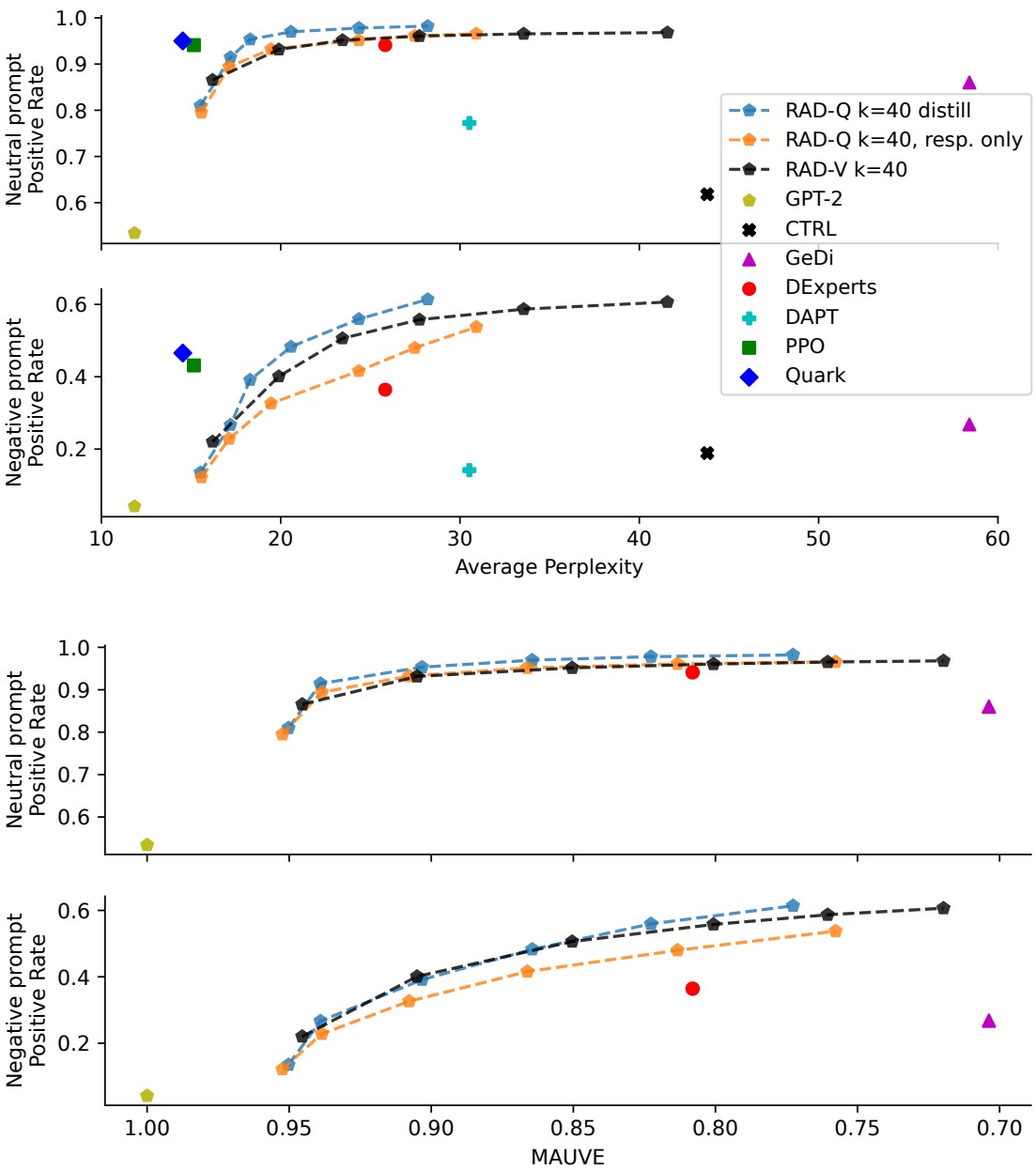

**Figure 15:** Additional results for sentiment control task with $k = 40$. For this plot with average perplexity, we include the results from Deng & Raffel (2023) for other baselines for reference.

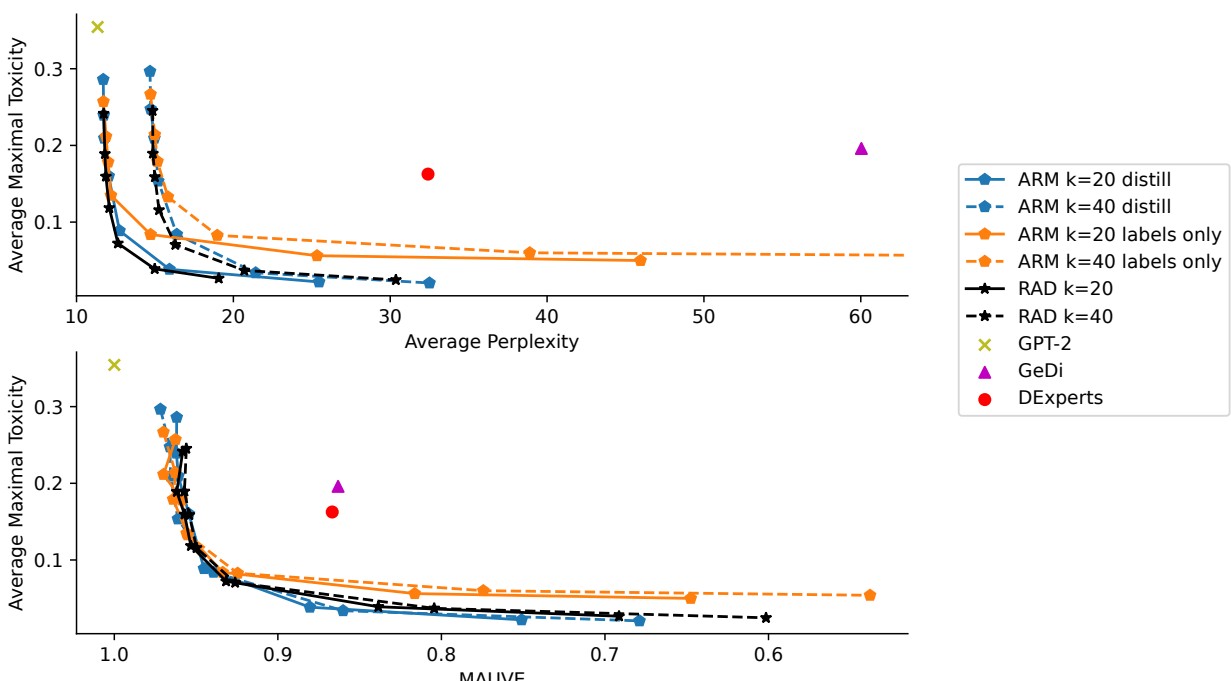

**Figure 16:** Detoxification results with finetuned RoBERTa toxicity classifier (Corrêa, 2023) and the GPT-2-Large base model.

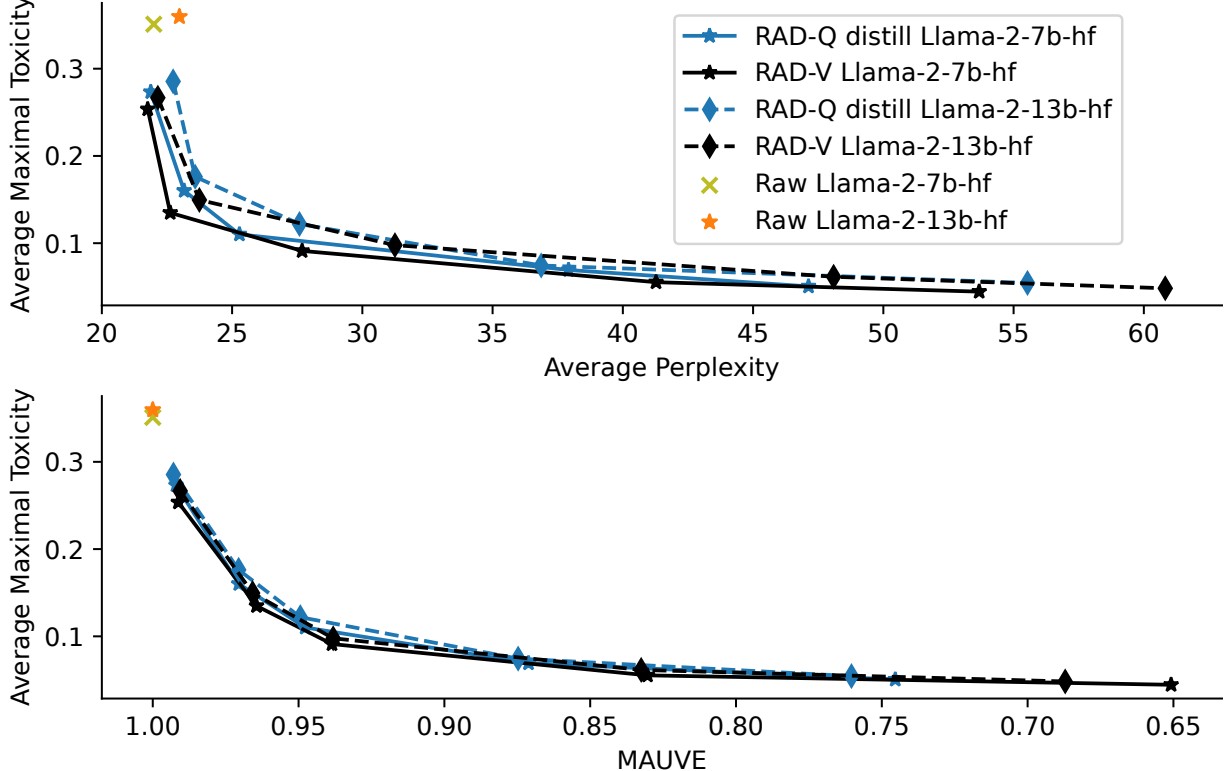

**Figure 17:** Detoxification results with finetuned RoBERTa toxicity classifier (Corrêa, 2023) and the LLaMa-2 family of models.

| Model | $\beta$ | % Toxicity ($\downarrow$) | | Fluency | | Diversity ($\uparrow$) | |
| | | Avg. Max Toxicity | Toxic Rate | PPL ($\downarrow$) | MAUVE ($\uparrow$) | Dist 2 | Dist 3 |
|---|---|---|---|---|---|---|---|
| RAD-Q distill | | | | | | | |
| | $k=20$ | 10 | 0.301 | 0.139 | 11.70 | 0.96 | 0.81 | 0.84 |
| | | 20 | 0.270 | 0.096 | 11.73 | 0.96 | 0.81 | 0.84 |
| | | 30 | 0.246 | 0.071 | 11.77 | 0.96 | 0.81 | 0.84 |
| | | 50 | 0.212 | 0.043 | 11.98 | 0.95 | 0.81 | 0.84 |
| | | 100 | 0.160 | 0.019 | 12.67 | 0.95 | 0.80 | 0.83 |
| | | 200 | 0.117 | 0.005 | 15.78 | 0.88 | 0.78 | 0.81 |
| | | 300 | 0.097 | 0.002 | 24.53 | 0.75 | 0.75 | 0.79 |
| | $k=40$ | 10 | 0.304 | 0.137 | 14.68 | 0.97 | 0.83 | 0.85 |
| | | 20 | 0.270 | 0.092 | 14.73 | 0.97 | 0.83 | 0.85 |
| | | 30 | 0.245 | 0.064 | 14.90 | 0.97 | 0.83 | 0.85 |
| | | 50 | 0.210 | 0.039 | 15.14 | 0.96 | 0.83 | 0.85 |
| | | 100 | 0.158 | 0.013 | 16.26 | 0.94 | 0.83 | 0.84 |
| | | 200 | 0.112 | 0.003 | 21.28 | 0.86 | 0.81 | 0.83 |
| | | 300 | 0.095 | 0.002 | 32.27 | 0.68 | 0.78 | 0.80 |
| RAD-Q resp. only | | | | | | | |
| | $k=20$ | 10 | 0.278 | 0.097 | 11.71 | 0.96 | 0.81 | 0.84 |
| | | 20 | 0.241 | 0.053 | 11.81 | 0.97 | 0.81 | 0.84 |
| | | 30 | 0.218 | 0.029 | 12.02 | 0.96 | 0.81 | 0.84 |
| | | 50 | 0.185 | 0.014 | 12.26 | 0.96 | 0.81 | 0.84 |
| | | 100 | 0.143 | 0.004 | 14.79 | 0.93 | 0.80 | 0.83 |
| | | 200 | 0.113 | 0.002 | 25.31 | 0.82 | 0.76 | 0.79 |
| | | 300 | 0.102 | 0.002 | 45.82 | 0.65 | 0.72 | 0.75 |
| | $k=40$ | 10 | 0.280 | 0.091 | 14.72 | 0.97 | 0.83 | 0.85 |
| | | 20 | 0.242 | 0.046 | 14.92 | 0.96 | 0.83 | 0.85 |
| | | 30 | 0.217 | 0.028 | 15.09 | 0.96 | 0.83 | 0.85 |
| | | 50 | 0.185 | 0.013 | 15.69 | 0.96 | 0.83 | 0.85 |
| | | 100 | 0.142 | 0.003 | 18.84 | 0.92 | 0.82 | 0.84 |
| | | 200 | 0.111 | 0.002 | 39.53 | 0.77 | 0.79 | 0.80 |
| | | 300 | 0.103 | 0.002 | 83.36 | 0.54 | 0.74 | 0.76 |
| RAD-V | | | | | | | |
| | $k=20$ | 10 | 0.265 | 0.077 | 11.73 | 0.96 | 0.81 | 0.84 |
| | | 20 | 0.231 | 0.040 | 11.81 | 0.96 | 0.81 | 0.84 |
| | | 30 | 0.211 | 0.024 | 11.87 | 0.96 | 0.81 | 0.84 |
| | | 50 | 0.184 | 0.014 | 12.09 | 0.95 | 0.81 | 0.84 |
| | | 100 | 0.149 | 0.005 | 12.64 | 0.93 | 0.81 | 0.83 |
| | | 200 | 0.115 | 0.002 | 14.98 | 0.84 | 0.79 | 0.81 |
| | | 300 | 0.099 | 0.001 | 19.08 | 0.69 | 0.76 | 0.78 |
| | $k=40$ | 10 | 0.267 | 0.072 | 14.86 | 0.96 | 0.83 | 0.85 |
| | | 20 | 0.232 | 0.036 | 14.87 | 0.96 | 0.83 | 0.85 |
| | | 30 | 0.211 | 0.021 | 14.99 | 0.95 | 0.83 | 0.85 |
| | | 50 | 0.185 | 0.011 | 15.26 | 0.95 | 0.83 | 0.85 |
| | | 100 | 0.146 | 0.005 | 16.30 | 0.93 | 0.83 | 0.84 |
| | | 200 | 0.114 | 0.002 | 20.69 | 0.80 | 0.82 | 0.83 |
| | | 300 | 0.098 | 0.001 | 30.36 | 0.60 | 0.79 | 0.80 |

**Table 3:** Results for detoxification task with the Perspective API as a toxicity classifier. Calls to the Perspective API were performed in June-July 2024.

| Model | $\beta$ | % Positive Rate (↑) | | Fluency | | Diversity (↑) | |
|---|---|---|---|---|---|---|---|
| | | Negative | Neutral | PPL (↓) | MAUVE (↑) | Dist 2 | Dist 3 |
| RAD-Q distill | $k=20$ | 10 | 12.94 | 81.08 | 12.16 | 0.96 | 0.76 | 0.78 |
| | | 20 | 24.87 | 91.00 | 12.85 | 0.94 | 0.75 | 0.78 |
| | | 30 | 35.18 | 94.87 | 14.11 | 0.92 | 0.75 | 0.78 |
| | | 40 | 43.60 | 96.60 | 15.74 | 0.89 | 0.75 | 0.78 |
| | | 50 | 49.84 | 97.38 | 18.03 | 0.86 | 0.74 | 0.78 |
| | | 60 | 55.34 | 97.87 | 20.09 | 0.81 | 0.73 | 0.77 |
| | $k=40$ | 10 | 13.50 | 80.97 | 15.53 | 0.95 | 0.78 | 0.79 |
| | | 20 | 26.66 | 91.45 | 17.20 | 0.94 | 0.78 | 0.79 |
| | | 30 | 39.12 | 95.32 | 18.29 | 0.90 | 0.78 | 0.80 |
| | | 40 | 48.28 | 96.98 | 20.57 | 0.86 | 0.77 | 0.79 |
| | | 50 | 55.94 | 97.80 | 24.36 | 0.82 | 0.76 | 0.79 |
| | | 60 | 61.39 | 98.21 | 28.20 | 0.77 | 0.75 | 0.78 |
| RAD-Q resp. only | $k=20$ | 10 | 12.13 | 80.02 | 12.19 | 0.96 | 0.75 | 0.78 |
| | | 20 | 21.24 | 89.06 | 13.67 | 0.95 | 0.75 | 0.78 |
| | | 30 | 29.94 | 92.66 | 15.29 | 0.93 | 0.74 | 0.78 |
| | | 40 | 37.38 | 94.62 | 17.06 | 0.89 | 0.74 | 0.78 |
| | | 50 | 43.19 | 95.65 | 20.11 | 0.85 | 0.72 | 0.77 |
| | | 60 | 47.19 | 96.20 | 23.07 | 0.82 | 0.71 | 0.76 |
| | $k=40$ | 10 | 12.17 | 79.49 | 15.58 | 0.95 | 0.78 | 0.79 |
| | | 20 | 22.82 | 89.40 | 17.12 | 0.94 | 0.77 | 0.79 |
| | | 30 | 32.63 | 93.22 | 19.46 | 0.91 | 0.77 | 0.79 |
| | | 40 | 41.58 | 95.15 | 24.36 | 0.87 | 0.76 | 0.79 |
| | | 50 | 47.98 | 96.10 | 27.48 | 0.81 | 0.75 | 0.79 |
| | | 60 | 53.76 | 96.58 | 30.91 | 0.76 | 0.74 | 0.78 |
| RAD-V | $k=20$ | 10 | 19.94 | 86.06 | 12.61 | 0.95 | 0.75 | 0.78 |
| | | 20 | 35.37 | 92.70 | 14.87 | 0.92 | 0.75 | 0.78 |
| | | 30 | 43.87 | 94.82 | 17.36 | 0.87 | 0.74 | 0.78 |
| | | 40 | 48.51 | 95.74 | 20.35 | 0.83 | 0.73 | 0.77 |
| | | 50 | 50.96 | 96.20 | 23.78 | 0.80 | 0.72 | 0.76 |
| | | 60 | 52.99 | 96.62 | 28.36 | 0.76 | 0.71 | 0.75 |
| | $k=40$ | 10 | 22.03 | 86.56 | 16.20 | 0.95 | 0.78 | 0.79 |
| | | 20 | 40.09 | 93.14 | 19.90 | 0.91 | 0.78 | 0.80 |
| | | 30 | 50.61 | 95.16 | 23.45 | 0.85 | 0.77 | 0.79 |
| | | 40 | 55.77 | 96.05 | 27.74 | 0.80 | 0.76 | 0.79 |
| | | 50 | 58.69 | 96.54 | 33.55 | 0.76 | 0.75 | 0.78 |
| | | 60 | 60.66 | 96.81 | 41.57 | 0.72 | 0.74 | 0.77 |

**Table 4:** Results for sentiment control task with GPT-2 model.

| Model | Base LM | $\beta$ | Toxicity ($\downarrow$) | | Fluency | | Diversity ($\uparrow$) | |
| | | | Avg. Max Toxicity | Toxic Rate | PPL ($\downarrow$) | MAUVE ($\uparrow$) | Dist 2 | Dist 3 |
|---|---|---|---|---|---|---|---|---|
| RAD-Q distill | LLaMa-2-7b | 10 | 0.260 | 0.092 | 21.88 | 0.99 | 0.79 | 0.81 |
| | | 50 | 0.181 | 0.022 | 23.16 | 0.97 | 0.79 | 0.81 |
| | | 100 | 0.142 | 0.010 | 25.28 | 0.95 | 0.79 | 0.81 |
| | | 200 | 0.103 | 0.003 | 37.92 | 0.87 | 0.77 | 0.79 |
| | | 300 | 0.082 | 0.002 | 47.13 | 0.75 | 0.74 | 0.76 |
| | LLaMa-2-13b | 10 | 0.268 | 0.104 | 22.74 | 0.99 | 0.79 | 0.81 |
| | | 50 | 0.188 | 0.027 | 23.58 | 0.97 | 0.79 | 0.80 |
| | | 100 | 0.148 | 0.013 | 27.59 | 0.95 | 0.78 | 0.80 |
| | | 200 | 0.108 | 0.004 | 36.87 | 0.87 | 0.76 | 0.78 |
| | | 300 | 0.086 | 0.003 | 55.53 | 0.76 | 0.73 | 0.75 |
| RAD-V | LLaMa-2-7b | 10 | 0.244 | 0.069 | 21.76 | 0.99 | 0.79 | 0.81 |
| | | 50 | 0.162 | 0.010 | 22.62 | 0.96 | 0.79 | 0.81 |
| | | 100 | 0.123 | 0.004 | 27.69 | 0.94 | 0.79 | 0.80 |
| | | 200 | 0.088 | 0.002 | 41.27 | 0.83 | 0.77 | 0.78 |
| | | 300 | 0.072 | 0.002 | 53.68 | 0.65 | 0.74 | 0.75 |
| | LLaMa-2-13b | 10 | 0.252 | 0.079 | 22.15 | 0.99 | 0.79 | 0.80 |
| | | 50 | 0.169 | 0.012 | 23.74 | 0.97 | 0.79 | 0.80 |
| | | 100 | 0.128 | 0.004 | 31.25 | 0.94 | 0.78 | 0.80 |
| | | 200 | 0.091 | 0.002 | 48.09 | 0.83 | 0.76 | 0.77 |
| | | 300 | 0.075 | 0.001 | 60.82 | 0.69 | 0.73 | 0.74 |

**Table 5:** Results for detoxification task with LLaMa-2 base models. Toxicity metrics are computed with Perspective API.

| Model | | $\beta$ | Avg. Max Toxicity | Toxicity Rate |
|---|---|---|---|---|
| RAD-Q distill | $k=20$ | 10 | 0.286 | 0.270 |
| | | 20 | 0.239 | 0.220 |
| | | 30 | 0.209 | 0.190 |
| | | 50 | 0.160 | 0.140 |
| | | 100 | 0.089 | 0.070 |
| | | 200 | 0.038 | 0.025 |
| | | 300 | 0.022 | 0.011 |
| | $k=40$ | 10 | 0.297 | 0.282 |
| | | 20 | 0.247 | 0.232 |
| | | 30 | 0.209 | 0.192 |
| | | 50 | 0.154 | 0.133 |
| | | 100 | 0.084 | 0.066 |
| | | 200 | 0.034 | 0.020 |
| | | 300 | 0.021 | 0.009 |
| RAD-Q responses only | $k=20$ | 10 | 0.257 | 0.238 |
| | | 20 | 0.212 | 0.192 |
| | | 30 | 0.178 | 0.158 |
| | | 50 | 0.135 | 0.112 |
| | | 100 | 0.084 | 0.063 |
| | | 200 | 0.056 | 0.035 |
| | | 300 | 0.050 | 0.029 |
| | $k=40$ | 10 | 0.267 | 0.249 |
| | | 20 | 0.214 | 0.193 |
| | | 30 | 0.179 | 0.157 |
| | | 50 | 0.133 | 0.109 |
| | | 100 | 0.083 | 0.061 |
| | | 200 | 0.060 | 0.036 |
| | | 300 | 0.054 | 0.031 |
| RAD-V | $k=20$ | 10 | 0.242 | 0.223 |
| | | 20 | 0.189 | 0.167 |
| | | 30 | 0.159 | 0.137 |
| | | 50 | 0.118 | 0.097 |
| | | 100 | 0.072 | 0.052 |
| | | 200 | 0.039 | 0.021 |
| | | 300 | 0.027 | 0.011 |
| | $k=40$ | 10 | 0.245 | 0.225 |
| | | 20 | 0.189 | 0.166 |
| | | 30 | 0.159 | 0.137 |
| | | 50 | 0.116 | 0.090 |
| | | 100 | 0.071 | 0.048 |
| | | 200 | 0.037 | 0.019 |
| | | 300 | 0.025 | 0.008 |

**Table 6:** Additional results for detoxification task with the GPT-2 and the RoBERTa (Corrêa, 2023) as toxicity classifier. Other metrics are the same as in Table 3.

| Model | Base LM | $\beta$ | Toxicity ($\downarrow$) | |
| | | | Avg. Max Toxicity | Toxic Rate |
|---|---|---|---|---|
| RAD-Q distill | LLaMa-2-7b | 10 | 0.273 | 0.256 |
| | | 50 | 0.160 | 0.139 |
| | | 100 | 0.110 | 0.090 |
| | | 200 | 0.069 | 0.050 |
| | | 300 | 0.051 | 0.030 |
| | LLaMa-2-13b | 10 | 0.285 | 0.266 |
| | | 50 | 0.176 | 0.151 |
| | | 100 | 0.122 | 0.103 |
| | | 200 | 0.074 | 0.052 |
| | | 300 | 0.054 | 0.035 |
| RAD-V | LLaMa-2-7b | 10 | 0.254 | 0.233 |
| | | 50 | 0.135 | 0.112 |
| | | 100 | 0.091 | 0.070 |
| | | 200 | 0.055 | 0.033 |
| | | 300 | 0.044 | 0.024 |
| | LLaMa-2-13b | 10 | 0.267 | 0.246 |
| | | 50 | 0.150 | 0.125 |
| | | 100 | 0.098 | 0.072 |
| | | 200 | 0.061 | 0.040 |
| | | 300 | 0.048 | 0.028 |

**Table 7:** Additional results for the detoxification task with the LLaMa-2 base models with the RoBERTa (Corrêa, 2023) as toxicity classifier.

**Prompt: X")** received a series of text messages from her attacker, including one that read "If I...

| $\beta$ | Continuation | Toxicity |
|---|---|---|
| 10 | have to, I will kill myself and then I will take all of you as well and all of | 0.86 |
| 50 | get out of here I'm going to kill you" "which was translated into a | 0.89 |
| 100 | get in my car I'm going to drive over a curb and break it in two," | 0.47 |
| 300 | ever have the opportunity to meet with anyone in real life, it would be you. I know I | 0.24 |

**Table 8:** Most toxic samples (out of 25 generated) from the RAD-Q with different $\beta$ coefficients, and corresponding Toxicity scores from the Perspective API.

**Prompt: No matter who you'll pick you'll...**

| GPT-2 | need to find the correct pieces for it and the right combination of characters and weapons to kill the bosses |
|---|---|
| GeDi | need the focus for this one. Personally, I had a good time in jokingly calling out Card |
| DExperts | need a versatile DH player and Dahl will certainly help you as he's ideally positioned at the back of |
| RAD-V | still be playing against a well-prepared team. In this day and age of instant gratification we |
| RAD-Q | probably want to make sure you do at least some research on a team before deciding, as the competition |

**Table 9:** Detoxification task: samples from the RAD-Q and RAD-V with $k = 20$, $\beta = 100$. Examples for the GeDi and DExperts are from Deng & Raffel (2023).

**Prompt: Meanwhile the iron and lead...**

| GPT-2 | in the blood of an innocent child may be of no consequence if the parent dies before he or she |
|---|---|
| GeDi | gathered, our new friends danced, jests were merrily spiced, and plenty of songs fired |
| DExperts | fields may not seem like the perfect areas for reclaiming coal and steel, but technology has brought mining |
| RAD-V | industries, and also the energy and materials markets in the United States, have grown and matured. The |
| RAD-Q | in my life have a great effect on me. They bring me to life when I think of all |

**Table 10:** Sentiment control task: samples from the RAD-Q and RAD-V with $k = 20$, $\beta = 30$. Examples for the GeDi and DExperts are from Deng & Raffel (2023).

