# OpenReview forum: "On the Low-Rank Parametrization of Reward Models for Controlled Language Generation"
_TMLR — Accepted by TMLR_

### Review · Reviewer_PXVR · 2025-05-01

**Summary Of Contributions:**

This paper introduces an analysis and practical implementation of low-rank parametrizations for reward models used in guided decoding of language models. The paper formalizes the expressiveness trade-off between two reward model styles --- V-style (token-conditioned) and Q-style (prefix-conditioned) --- and highlights that Q-style suffers from a “rank bottleneck” but can still suffice in practice. Methodologically, the paper proposes RAD-Q, a more efficient, low-rank variant of the RAD approach, achieving a single forward pass per decoding step (as opposed to RAD-V’s per-token cost), thus making decoding faster and more scalable. Empirically, experiments are conducted on detoxification and sentiment control tasks. Results demonstrate that RAD-Q performs comparably to or even better than RAD-V in both tasks, especially when trained via distillation.

**Audience:**

Yes

**Claims And Evidence:**

Yes

**Requested Changes:**

Elaborating more on the link between V/Q-learning and reward model learning (cf. Introduction section, page 1) could be useful. In Figure 1, additional illustrations on the plot units can be useful.

Can the authors discuss more about their empirical insights in implementing the methods? --- Since the distillation method is a two-stage implementation, a detailed discussion on the engineering side can be helpful for the paper's readers to quickly adapt the method to their tasks.


Minor:

typo in Figure 4 Caption: distil -> distill

I would recommend including the LLaMA results in the main text.

**Strengths And Weaknesses:**

Strength:

The paper provides an interesting matrix completion perspective on reward modeling, including formal definitions of minimal rank and ε-rank, clarifying the practical expressivity of reward models.

The experiments in this paper span multiple base models, two tasks, multiple metrics, and plenty of insightful ablation studies.

Weaknesses:

This is not actually a limitation of this specific paper, but I wonder if the authors could elaborate more on the assumptions of low-rank and full-rank properties of different tasks. I'm aware this might be out of the scope of the current paper --- but would it be possible to control the rank in some synthetic setups and demonstrate the RAD-V/Q's performance against LLM fine-tuning methods?

The RAD-V - RAD-Q distillation may introduce an extra computational burden.

---

> ### Author Response · Authors · 2025-06-20
> **Authors response**
>
> We thank the reviewer for their feedback. We will incorporate your suggestions and we address your concerns bellow.
>
> > This is not actually a limitation of this specific paper, but I wonder if the authors could elaborate more on the assumptions of low-rank and full-rank properties of different tasks. I'm aware this might be out of the scope of the current paper --- but would it be possible to control the rank in some synthetic setups and demonstrate the RAD-V/Q's performance against LLM fine-tuning methods?
>
> We think this idea is very interesting and may shed light on limitations of LLM finetuning too i.e. synthetic datasets may be unlearnable by LLMs exactly if the rank exceeds the model dimension. This seems like a great idea to pursue in a subsequent project.
>
> > The RAD-V - RAD-Q distillation may introduce an extra computational burden.
>
>
> The distillation step indeed introduces additional computational overhead, where we need to run an additional forward pass through the RAD-V model and maintain two models in memory. Interestingly, Figure 8 indicates that the distilled model has an even lower rank, which can potentially provide insights into training methods that can bypass the distillation step. We want to emphasize that a RAD-Q model trained from scratch (no distillation) is not drastically worse compared to the distilled RAD-Q.
>
> > Can the authors discuss more about their empirical insights in implementing the methods?
>
> We will provide the necessary code, and we provide the implementation details in Appendix E (Training Details).
>
>
> > Elaborating more on the link between V/Q-learning and reward model learning (cf. Introduction section, page 1) could be useful.
>
> We will add the additional references and clarifications regarding the reward modeling to the introduction.
>
> > In Figure 1, additional illustrations on the plot units can be useful.
>
> We updated the Figure 1 illustration caption by clarifying the units and we fixed the typos in the text.
>
>
> > I would recommend including the LLaMA results in the main text.
>
> We agree this is great suggestion. We moved Figure 15 plot with a LLaMa result to the main part. The changes may slightly exceed 12pg, so we kindly ask the reviewers and AE to confirm if this is ok in the final decision.

---

### Review · Reviewer_9byQ · 2025-05-19

**Summary Of Contributions:**

The paper studies the use of reward models for controlled generation in language models (LLMs). Specifically, the paper proposes a low-rank parameterization of reward models that leads to an efficient reward augmented decoding (RAD) method. Additionally, the paper aims to show that the existing RAD methods are not necessarily low-rank and that there may exist reward matrices that are high rank by construction. The proposed method, RAD-Q, is empirically shown to work reasonably well on detoxification and sentiment control datasets.

**Audience:**

Yes

**Broader Impact Concerns:**

This paper proposes an approach to improve controlled generation. The paper would definitely benefit from having a Broader Impact Statement in its content.

**Claims And Evidence:**

No

**Requested Changes:**

- Please see question sunder weaknesses, summarized below:
  - Would the paper read better if the order of presentation is changed? Present RAD-Q right away so there is no confusion on this method being the main contribution
  - Are there cases useful in practice that shows full-rank reward matrix
  - Is effective rank a better measure of matrix rank for the analysis conducted in the paper?

- There appears to be a typo on Page 3 where-in the papers cited for GeDI and DExperts are swapped. If this is the case, please fix.

- In Figure 2, the use of "d" to compare to numerical rank is confusing. Does "d" have to be displayed if the reward matrix is of size N by |V|?

**Strengths And Weaknesses:**

## Strengths

- The paper provides a method that may be an efficient alternative to an established reward-augmented decoding approach (RAD-V). The connection made to matrix-sensing literature in terms of parameterizing the reward matrix may be useful to the theory folks working on controllable generation.

- Empirical results on sentiment control and detoxification suggest that the proposed RAD-Q method is competitive with baseline RAD (RAD-V) method.

## Weaknesses

- The paper goes over why one may not want to always assume reward models can be low-rank. However, the paper's presentation currently delays the reader from getting to the core contribution which is RAD-Q which maybe confusing to the reader. Changing up the order of presentation could help improve readability

- Related to above, it is not clear to the reader whether the argument about high-rank vs low-rank models is even relevant in practice. On the one hand, by construction the paper provides an example of full-rank reward matrix but the empirical results suggest that low-rank reward matrices are a reasonable assumption. Are there practical use cases where low-rank parameterization does not work well?

- The approach used in the paper to calculate matrix rank is a concern. Have you the authors considered alternative approach like effective rank [1]? My concern is the rank of a matrix is heavily dependent on the underlying SVD implementation which can change over time and also across precisions. Perhaps a measure like effective rank might be better?

[1] Roy & Vetterli https://www.eurasip.org/Proceedings/Eusipco/Eusipco2007/Papers/a5p-h05.pdf

---

> ### Author Response · Authors · 2025-06-20
> **Authors response**
>
> We thank the reviewer for their feedback. We will incorporate your suggestions and we address your concerns bellow.
>
> >Would the paper read better if the order of presentation is changed? Present RAD-Q right away so there is no confusion on this method being the main contribution
>
> We will revisit the paper structure and clarify through emphasis or reordering.
>
> > it is not clear to the reader whether the argument about high-rank vs low-rank models is even relevant in practice. Are there cases useful in practice that shows full-rank reward matrix
>
>
> From our analysis of RAD, it seems that the low-rank phemonenon is likely to be observed, in part due to the sparisity of the reward matrix. We agree with the reviewer that it would be interesting to show real-life examples of high-rank phenomena in addtion to the construction we provide in the paper.
>
> We think it is realistic to observe a high rank phenomena, in particular, when rewards are highly context dependent. Consider an application of a LM to the maths tasks, where you have examples in form of "x + y = z", where x, y, z are integer number tokens e.g. in V=[0..999]. Let us define a reward function as R("x + y = z") = 1 if the expression is true, 0 if false. Then by considering all possible contexts "x + y =", it is easy to show that the reward matrix is full rank since the rows of the reward matrix include all one-hot vectors representing the correct answer.
>
> > The approach used in the paper to calculate matrix rank is a concern. Have you the authors considered alternative approach like effective rank [1]? My concern is the rank of a matrix is heavily dependent on the underlying SVD implementation which can change over time and also across precisions. Perhaps a measure like effective rank might be better?
> [1] Roy & Vetterli https://www.eurasip.org/Proceedings/Eusipco/Eusipco2007/Papers/a5p-h05.pdf
> > Is effective rank a better measure of matrix rank for the analysis conducted in the paper?
>
>
> To calculate the matrix rank, we use a standard and stable SVD implementation (for SVD, the Numpy implementation calls the LAPACK routine, as follows from Numpy documentation for SVD). Effective rank (Shannon entropy of the singular values) does not seem to be a better alterative compared to standard matrix rank implementation: 1) effective rank is less interpretable, 2) it is hard to compare effective rank to the model dimension, 3) effective rank computation would probably also require SVD to obtain singular values. We will discuss this and several other related approaches for rank calculation in the paper.
>
>
> > There appears to be a typo on Page 3 where-in the papers cited for GeDI and DExperts are swapped. If this is the case, please fix.
>
> We fixed the typos in the text.
>
>
> > In Figure 2, the use of "d" to compare to numerical rank is confusing. Does "d" have to be displayed if the reward matrix is of size N by |V|?
>
> Figure 2 shows the rank of a RAD-V model, and $d$ here is only for reference, and is more relevant when considering to train a RAD-Q model. When considering a RAD-Q approach e.g. distilling RAD-V into RAD-Q, it is important to ask whether the rank capacity of RAD-Q model is enough to model RAD-V outputs: whether model dimension is higher than the rank). We will clarify the connection in the caption of Figure 2.

---

> > ### Comment · Reviewer_9byQ · 2025-07-03
> > **Response to rebuttal**
> >
> > I would like to thank the authors for their rebuttal and response. It would be good to see an updated draft with changes applied to the paper but I will leave this to the discretion of the authors and AE.
> >
> > > To calculate the matrix rank, we use a standard and stable SVD implementation (for SVD, the Numpy implementation calls the LAPACK routine, as follows from Numpy documentation for SVD). Effective rank (Shannon entropy of the singular values) does not seem to be a better alterative compared to standard matrix rank implementation: 1) effective rank is less interpretable, 2) it is hard to compare effective rank to the model dimension, 3) effective rank computation would probably also require SVD to obtain singular values. We will discuss this and several other related approaches for rank calculation in the paper.
> >
> > The authors are indeed correct that effective rank requires SVD values as well. My concern with matrix rank is related to the "tol" parameter value whose default value can be found in a library's documentation. The documentation page also states how the default value may be set by the user based on their needs. My goal in pointing to other measures is to avoid having this implementation brittleness. With that said, I understand the authors' point of view and am satisfied with the response.

---

### Review · Reviewer_Jz7g · 2025-06-12

**Summary Of Contributions:**

This is an interesting paper exploring the difference between a language-modeling (Q) style parameterization (passing in entire vocabulary) of reward models and the typical parameterization of reward models (V) which involves needing a full forward pass for any individual token. To summarize my understanding of the paper:
1. It first provides some background on different approaches related to controlled decoding and how they can fit within a dichotomy of Q-style vs V-style reward models. This dichotomy is not entirely new however my understanding is that this paper is an early paper taking this dichotomy into mind more seriously.
2. It then provides a novel mathematical framework for thinking about the task and representational complexity of reward models using a matrix-based framework where, if I roughly understand correctly, rows represent different possible prefixes, and columns represent possible continuation tokens, and the score represents some expected reward when continuing from the prefix given the new token. They then discuss things in terms of the rank of these matrices for both reward formulations.
3. My understanding is that their central set of claims are that (a) counter to intuition, V-style reward models don’t actually tend to learn expressive high-rank matrices, possibly due to data sampling biases and also tasks maybe being too simple to need high-rank reward matrices, and (b) due to softmax bottleneck Q-style reward models are fundamentally limited to lower reward rank, but that this may be ok given (a). There’s additional discussion of how the structure of data may affect these conditions.
4. They then present some experiments on fairly simplistic sentiment classification and toxicity tasks which somewhat validate the above claims, in addition to showing some small gains with a novel distillation approach from V to Q rewards. Likewise there’s some interesting analysis and evidence suggesting a decoupling between rank of learned representations and quality/control tradeoffs.
My understanding is that the primary contributions here are the framing and analysis of reward through the idea of rank matrices, and a stronger case for Q-style reward models compared to initial prior work in that area.

**Audience:**

Yes

**Claims And Evidence:**

Yes

**Requested Changes:**

Suggested Changes:
Major: I think any sort of experiment examining a more realistic math or chat setting (e.g. using ARMoRM) would really improve this paper. To understand (a) cases where the task requires higher rank to satisfy (b) understand whether the assumption that Q-style and V-style rewards can function similarly in complex reward modeling tasks used in practice. The paper is currently a bit more cautious in its claims which I appreciate, but I think this investigation shouldn't be too difficult to run and could allow the paper to make stronger claims that would make the paper much stronger.
Minor: This paper looks primarily at RAD, which is one way of controlling generation with reward models, however it could be good to understand other algorithms for reward-guided decoding, such as a simple best-of-n approach. I understand this is more computationally expensive so it’d mainly helpful context
Minor: Motivation for improving cost-effectiveness can either be to (a) reduce current overhead or (b) allow greater scaling of inference-time compute. I’d be curious how the rank tradeoffs here affect for example best-of-n over candidates sampled from RAD-conditioned distribution (e.g. do these findings apply or change when we’re trying to scale up inference time compute), again in particular for more challenging control tasks or rewards

[1] ARMORM -> Interpretable Preferences via Multi-Objective Reward Modeling and Mixture-of-Experts
[2] RewardBench -> RewardBench: Evaluating Reward Models for Language Modeling

**Strengths And Weaknesses:**

Strengths:
- I really like the framework proposed here for thinking about task complexity in terms of rank over data, it’s definitely very fresh in the context of reward modeling, and the framing and connection to efficiency is unique. I’d in fact think that this framework has broader implications and applications for language modeling even outside the context of reward-guided decoding
- The experiments generally seem reasonable, and reveal some interesting patterns (such as lower rank sometimes not affecting quality so much with distillation) that I think will be useful for further exploration.
- I appreciate the usage of example figures to walk through some of the key ideas (e.g. Figure 3)
- There’s a good number of baselines
- The inclusion of a comparison of different distillation approaches is interesting, though maybe not analyzed as much

Weaknesses:
- A very serious limitation is that this work doesn’t seem connected to a lot of the empirical reward modeling work being done at larger-scale realistic chat settings (e.g. RewardBench submissions such as ARMoRM) or math reasoning settings, which are the main context in which reward models are typically discussed. Given how simple sentiment and toxicity control are highly simple tasks (even in the older context of controllable generation work cited by this paper) I worry that these findings might be highly specific and not generally applicable, e.g. maybe most realistic control or reward functions actually do need higher rank reward functions. The paper acknowledges this but doesn’t dig into this empirically.
- The explanation of the idea of rank felt a little unclear to me, especially in the context of true reward, I think a figure to visualize the key ideas of the paper, and maybe a simple worked example (e.g. of reward and rank computation) w

---

> ### Author Response · Authors · 2025-06-20
> **Authors response**
>
> We thank the reviewer for their feedback. We will incorporate your suggestions and we address your concerns bellow.
>
> > The explanation of the idea of rank felt a little unclear to me, especially in the context of true reward, I think a figure to visualize the key ideas of the paper, and maybe a simple worked example (e.g. of reward and rank computation) w
>
> We consider reward modeling as a problem of approximating the future rewards. By enumerating the contexts, and by showing that minimization of squared loss for future rewards is equivalent to approximating the expected future reward (eq. 5), we define an |N|x|V| *reward matrix*, where each value represents the expected reward when choosing a next token given a context. We estimate the rank of the predicted reward (sub-)matrices directly; but to estimate the rank of the target reward matrices, since some values are unobserved given data, we define the *minimal rank* as the rank of the lowest-rank completion consistent with the observed matrix values (or minimal $\epsilon$-rank for a lowest-rank completion with a small error). We will revisit the 3.1.1 section for the final version to improve clarity.
>
>
> > Major: I think any sort of experiment examining a more realistic math or chat setting (e.g. using ARMoRM) would really improve this paper. To understand (a) cases where the task requires higher rank to satisfy (b) understand whether the assumption that Q-style and V-style rewards can function similarly in complex reward modeling tasks used in practice. The paper is currently a bit more cautious in its claims which I appreciate, but I think this investigation shouldn't be too difficult to run and could allow the paper to make stronger claims that would make the paper much stronger.
>
>
> Regarding the datasets used in our work (Jigsaw, sentiment), we emphsize that these are the datasets commonly used in controlled generation literature and realistic, though they might be different from those used for reward modeling in broader context.
>
> To perform a quick experiment on reward modeling datasets from ARMoRM, we analyse the minimal ranks of the target reward matrices obtained from these datasets.
> In particular, we have added an additional experiment (see Table 2 in Appendix C.3) showing results on two datasets used to train a reward model in ARMoRM paper: BeaverTails (safety) and HelpSteer (helpfulness).
>
> The results are similar to what we observed for Jigsaw/sentiment datasets, namely that a rank less than model dimention is enough to well-approximate the target reward matrix with $\text{MSE}<10^{-6}$. This finding provides additional evidence that it is possible and realistic to have rank-constrained models perform on par with unconstrained ones given that low-rank seems to be enough to well approximating the target reward matrices.
>
> > Minor: This paper looks primarily at RAD, which is one way of controlling generation with reward models, however it could be good to understand other algorithms for reward-guided decoding, such as a simple best-of-n approach. I understand this is more computationally expensive so it’d mainly helpful context
> > Minor: Motivation for improving cost-effectiveness can either be to (a) reduce current overhead or (b) allow greater scaling of inference-time compute. I’d be curious how the rank tradeoffs here affect for example best-of-n over candidates sampled from RAD-conditioned distribution (e.g. do these findings apply or change when we’re trying to scale up inference time compute), again in particular for more challenging control tasks or rewards
>
> We will add best-of-n sampling discussion to the related work, emphasizing that best-of-n sampling  might require large number of full hypotheses to form the hypothesis poll (Sun et al 2024), and it might have a high rejection rate in case all hypotheses lead to a low reward. Our work focuses on controlled decoding scenarios, where we guide *intermediate* solutions during decoding. Additionally, we note that for the best-of-n sampling we would only need a RAD-V-like reward model to score full hypotheses, meaning the V/Q comparison is not relevant to the best-of-n sampling scenario.
>
> - Sun et al 2024. Fast Best-of-N Decoding via Speculative Rejection

---

> > ### Comment · Reviewer_Jz7g · 2025-06-27
> > **Follow-up Response**
> >
> > Thanks for responding to my comments! I think the additional experiments definitely help strengthen the paper a lot and appreciate the extra clarifications. Am good to recommend this paper. I'd be excited to see future work thinking about the implications of this work more broadly in text generation even beyond reward modeling, or other possible extensions.

---

### Decision · Action_Editor_Vm1u · 2025-07-28

**Recommendation:** Accept as is

**Additional Comments:**

The reviewers all ended up positive about this paper. For my part, I think this paper does a good job of explaining the V vs. Q formulations of rewards here. It's an intuitive idea that is likely well understood already by many researchers working in this space, but the presentation here and the rank analysis brings something new beyond what I've seen in the past in the literature.

One comment raised by reviewer Jz7g, which I agree with, is that the tasks used in this paper are a bit "backward-facing." These were controlled generation tasks that featured prominently in literature in 2020-2024 but I'm not convinced they're where the "action" is going to be for future language models. This doesn't impact the intellectual rigor of the idea, and I think the results here are still compelling, but I will note that running with modern RMs will probably help adoption of these ideas going forward.

9byQ also brings up a point about the flow of the paper and the discussion of full-rank vs. low-rank. I agree that the narrative flow of the paper could perhaps be structured more to drive home the point that low-rank is sufficient rather than acknowledging that it depends.  However, I found the discussion of the pros and cons to be refreshing and illuminating about the conditions for full-rank vs. low-rank.  Of course, many higher-rank matrices in practice can be low-rank approximated well empirically, but I felt this point was clear enough.

**Audience:**

Yes

**Audience Explanation:**

Better understanding of reward models and this kind of guided decoding is a very timely topic

**Claims And Evidence:**

Yes

**Claims Explanation:**

This paper looks at expert-guided (reward model-guided) decoding of LLMs, and in particular the parameterization of that reward model. The paper claims to implement a low-rank version of that expert on detoxification and sentiment control tasks while incurring less computational expense. These claims are substantiated in the paper by both theoretical/conceptual discussion and empirical results, as all authors agree, and while more could always be done empirically, what's here is above the bar.